# Melt-quenched carboxylate metal–organic framework glasses

Minhyuk Kim [1], Hwa-Sub Lee[2], Dong-Hyun Seo[3], Sung June Cho [4] ✉, Eun-chae Jeon [2] ✉ & Hoi Ri Moon [5] ✉

Although carboxylate-based frameworks are commonly used architectures in metal-organic frameworks (MOFs), liquid/glass MOFs have thus far mainly been obtained from azole- or weakly coordinating ligand-based frameworks. This is because strong coordination bonds of carboxylate ligands to metals block the thermal vitrification pathways of carboxylate-based MOFs. In this study, we present the example of carboxylate-based melt-quenched MOF glasses comprising $Mg^{2+}$ or $Mn^{2+}$ with an aliphatic carboxylate ligand, adipate. These MOFs have a low melting temperature ($T_m$) of 284 °C and 238 °C, respectively, compared to zeolitic-imidazolate framework (ZIF) glasses, and superior mechanical properties in terms of hardness and elastic modulus. The low $T_m$ may be attributed to the flexibility and low symmetry of the aliphatic carboxylate ligand, which raises the entropy of fusion ($\Delta S_{fus}$), and the lack of crystal field stabilization energy on metal ions, reducing enthalpy of fusion ($\Delta H_{fus}$). This research will serve as a cornerstone for the integration of numerous carboxylate-based MOFs into MOF glasses.

Metal–organic frameworks (MOFs) are coordination networks with potential pores in a well-ordered structure composed of metal ions and polydentate organic ligands[1]. Over the past few decades, the field of MOFs has significantly expanded because of their high designability and tunability[2]. Despite their various properties, the practical applications of MOFs are limited because of their crystalline powder nature and low processability[3,4]. To overcome the limitations of crystalline MOFs, efforts have been made to transform MOFs into more practical and versatile forms and shapes through integration with polymers, pelletization, and processing into beads[5]. Among these approaches, meltable MOFs have recently gained significant attention, as their liquid phase allows for molding without dependence on other materials[6]. Moreover, molten MOFs can generate a distinct type of material, MOF glasses via a melt-quenching process[7,8]. These glass structures retain the components of the original crystal and exhibit unique properties such as a monolithic manner[9], enhanced ion conductivity[9], transparency[10], and luminescence[11]. They also have a distorted pore network distinct from the crystalline MOFs[12].

To enable melting in a MOF, the MOF must have either a low melting temperature ($T_m$) or a high thermal decomposition temperature ($T_d$) to satisfy the condition, $T_m < T_d$ (Fig. 1). This requirement arises from the fundamental concern that the average local coordination environment of the structures must be maintained while their long-range order is lost[13,14]. So far, studies on meltable MOFs have mostly focused on zeolitic-imidazolate frameworks (ZIFs) with high $T_d$ owing to their thermally stable azole ligands, and coordination polymers (CPs) composed of phosphates, amides, and sulfonates, which form weak coordination bonds with metals, thereby lowering the $T_m$ of the framework[15,16].

Despite recent advancements in the MOF glass field, an important area that still need attention is the melting and vitrification of carboxylate-based MOFs, which constitute a significant majority of

[1]Department of Chemistry, School of Natural Science, Ulsan National Institute of Science and Technology (UNIST), Ulsan 44919, Republic of Korea. [2]School of Materials Science and Engineering, University of Ulsan, 93 Daehak-ro, Nam-gu, Ulsan 44610, Republic of Korea. [3]Major of Nano-Mechatronics, University of Science and Technology, 217, Gajeong-ro, Yuseong-gu, Daejeon 34113, Republic of Korea. [4]Department of Chemical Engineering, Chonnam National University, 77 Yongbong-Ro, Buk-gu, Gwangju 61186, Republic of Korea. [5]Department of Chemistry and Nano Science, Ewha Womans University, Seoul 03760, Republic of Korea. ✉e-mail: sjcho@chonnam.ac.kr; jeonec@ulsan.ac.kr; hoirimoon@ewha.ac.kr

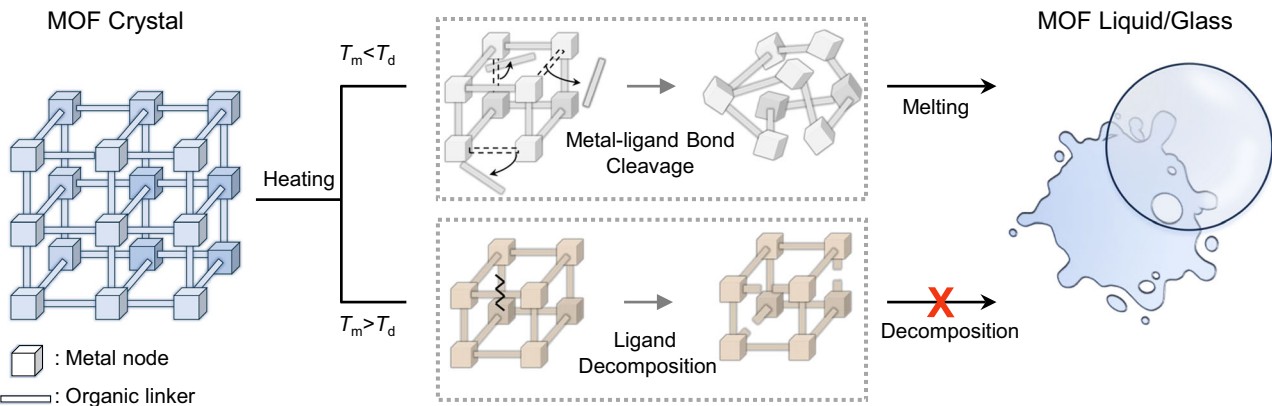

**Fig. 1 | Schematic graphics illustrating the principle of MOF melting.** For the MOF to melt, partial decoordination of the metal-ligand bonds have to occur before the thermal decomposition of framework ($T_m < T_d$). Therefore, a meltable MOF can be designed by controlling $T_m$ and $T_d$. The arrow with the red cross indicates that when a MOF has a higher $T_m$ than $T_d$, the melting process is forbidden during the heating process.

MOFs[17]. Most carboxylate-based MOFs decompose before vitrification owing to the strong bonds between carboxylate and metal centers, which elevate the $T_m$ of the framework above its $T_d$[15,18]. Extensive research has been conducted on carboxylate-based MOFs to date; however, to the best of our knowledge, no melting behavior has been reported among MOFs whose structures use purely carboxylates as ligands. Considering the active research on thermally decomposing MOFs to create nanomaterials, such as metal oxides and carbon, the absence of meltable carboxylate-based MOFs becomes even more surprising[19]. Given the recent discovery of the liquid and glassy states of MOFs, it is possible that earlier researchers in MOF deocomposition studies did not observe or recognize these states in carboxylate-based MOFs. Another possible factor is the prevalent strategies in reticular design. These strategies commonly involve using aromatic ligands to create well-ordered structures and metals with strong coordination bonds to form robust frameworks, resulting in higher crystallinity, porosity, and good stability in MOFs[20,21]. However, aromatic ligands have few conformations due to their rigid local structure and high symmetry properties, which do not provide entropic benefits for the melting of MOFs[13,22]. Furthermore, the strong metal-ligand bond directly contributes to increasing the MOF's $T_m$, thus hindering its melting.

We introduce meltable carboxylate-based MOFs consisting of $Mg^{2+}$ or $Mn^{2+}$ ions and an *aliphatic* carboxylate linker, adipate (adp, −(OOC)(CH_2)_4(COO)−). Compared to aromatic carboxylate ligands, aliphatic carboxylate ligands have lower thermal stability and a higher degree of conformational freedom. The $d^0$ and $d^5$ configuration of metals may lead to a reduction in the thermal energy required for the breaking of metal-ligand bonds, owing to their low crystal field stabilization energy (CFSE), which is similar to $d^{10}$ metals found in existing CP/ZIF glasses[7,8,10,15]. Additionally, the aliphatic carboxylate amplifies the gap in rotational entropy between the solid and liquid phases of MOFs compared to an aromatic linker, due to its ability to adopt numerous conformations in the liquid phase[15,22]. Simultaneously, the aliphatic linker may reduce the thermal stability of the framework. Based on these properties, we have previously demonstrated the thermal conversion of aliphatic ligand-based MOFs with low $T_d$ into hierarchically nanoporous metal oxides with nanocrystalline frameworks[23].

Recently, there have been a few reports on the synthesis of carboxylate-based MOF glasses[24–27]. However, these studies differ from the present work. The starting materials for glasses in these reports were derived from hydrogen-bonded networks of the metal complexes or from disorder-induced frameworks created through dehydration. Certainly, adopting these approaches is a compelling strategy for synthesizing MOF glasses. These methods not only offer a facile synthesis process with substantial industrial potential but also the capability to vitrify thermally unstable frameworks. Nevertheless, achieving a molten phase in MOFs remains a notable challenge due to its potential to unlock unprecedented applications distinct from non-meltable frameworks. The presence of a liquid phase in MOFs introduces captivating possibilities, including the potential for unconventional structural evolution[28], the creation of eutectic composites with other substances[29], and the utilization of MOFs as matrix materials themselves[30], contrasting with the conventional application of MOFs as solid additives[31]. Moreover, a fundamental understanding of meltable carboxylate-based MOFs may assist future studies aimed at imparting melt behavior to existing MOFs that have non-meltable properties.

While the existing approaches circumvent the thermodynamic challenges of carboxylate-based frameworks, the absence of a solid-liquid phase transition or $T_m$ in crystalline carboxylate frameworks restricts the variety of reported liquid/glass MOFs[27]. Consequently, it has impeded the establishment of rational design principles for meltable MOF structures.

Here, we utilize the low $T_m$ of the crystalline MOFs ([Mg_4(adipate)_4(DMA)(H_2O)] = C-Mg-adp and [Mn_2(adipate)_2(DMA)] = C-Mn-adp) by controlling the enthalpy of fusion ($\Delta H_{fus}$) and the entropy of fusion ($\Delta S_{fus}$) to trigger their thermal transition into the liquid phase, creating the carboxylate-based MOF glasses (G-Mg-adp and G-Mn-adp, respectively). These adipate MOFs exhibit a high glass-forming ability (GFA), signifying their capacity to easily vitrify in their liquid state while preventing recrystallization. X-ray total scattering data and pair distribution functions (PDFs) confirmed that G-Mg-adp retains the connectivity between the carboxylate and metal ions. The mechanical properties of G-Mg-adp were characterized using nanoindentation and exhibited higher hardness ($H$) and elastic modulus ($E$) than those of the reported CP glasses.

## Result

### Materials preparation and characterization

The solvothermal reaction of $Mg(NO_3)_2 \cdot 6H_2O$ and adipic acid in $N,N'$-dimethylacetamide (DMA) and methanol (MeOH) yielded block-shaped crystals of [Mg_4(adipate)_4(DMA)(H_2O)]·5DMA·2MeOH·4H_2O, C-Mg-adp[23]. In this structure, the coordination of carboxylate with $Mg^{2+}$ forms the secondary building units (SBUs) of 1D infinite Mg−O chains along the $b$-axis, which are interconnected by adp ligands to generate the 3D porous network (Fig. 2a). The building units serve as the SBUs, creating helical 1D Mg-adp chains. The thermal stability of C-Mg-adp was monitored using thermogravimetry analysis (TGA) with a ramp rate of 10 °C min$^{-1}$ under an inert atmosphere (Supplementary Fig. 1). The TGA trace reveals that after the initial weight loss corresponded

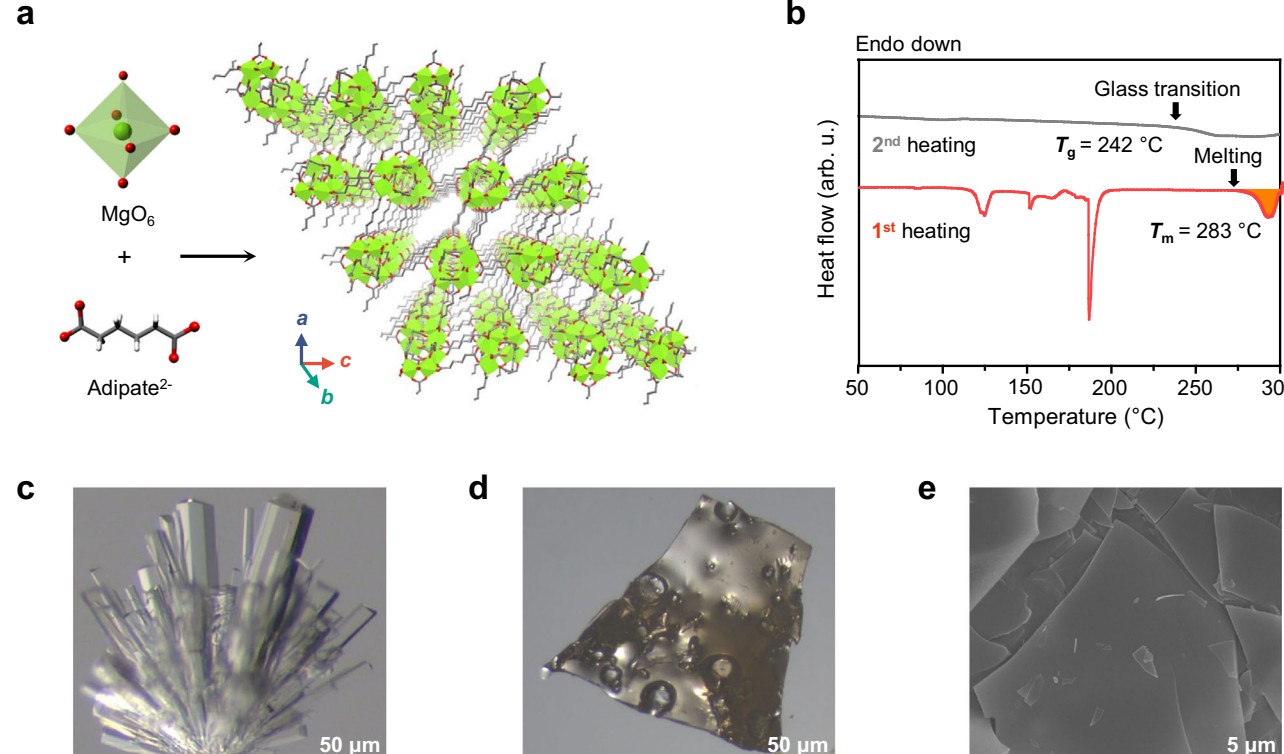

**Fig. 2 | Characterization of melt-quenching in *C*-Mg-adp crystals, and *G*-Mg-adp. a** Single-crystal X-ray structure of *C*-Mg-adp. Green, red, grey, and blue spheres represent Mg, O, C, and N atoms, respectively. Hydrogen atoms are omitted for clarity. **b** DSC curves of *C*-Mg-adp heated with a ramp rate of 10 °C min⁻¹ under argon, the red line corresponds to the first heating step, and the filled area indicates $\Delta H_{fus}$ for *C*-Mg-adp (134.5 kJ mol⁻¹). The grey line corresponds to the second heating step in the subsequent cycle under the same conditions. The $T_m$ and $T_g$ were evaluated by the onset point of each peak. **c, d** Optical microscope images of (**c**) *C*-Mg-adp and (**d**) *G*-Mg-adp, clearly showing the shape change of crystal by melt-quenching. **e** SEM image of *G*-Mg-adp with monolithic surface.

to the guest and coordinated molecules, there is a subsequent rapid weight loss indicating MOF decomposition occurred around 320 °C ($T_d$). Interestingly, differential scanning calorimetry (DSC) measurements for *C*-Mg-adp showed an endothermic peak before $T_d$, ranging from 283–300 °C (Fig. 2b), indicating the melting transition of *C*-Mg-adp. The $T_m$ of *C*-Mg-adp is 283 °C, which is higher than that of weakly coordinated networks but lower than that of ZIFs[14,31]. In the subsequent heating after cooling, the glass transition of melt-quenched Mg-adp (*G*-Mg-adp) was observed at 242 °C ($T_g$). The liquid fragility index (dynamical parameters) of *G*-Mg-adp calculated with $T_g$ obtained from various heating rates (10–30 °C min⁻¹) was 33, which implies that its flowing is hardly observed (Supplementary Fig. 2)[7]. The structural transformations of Mg-adp during melting were examined using X-ray powder diffraction (XRPD) and in-situ variation temperatures XRPD (VT-XRPD) (Supplementary Figs. 3 and 4). XRPD data shows that *C*-Mg-adp exhibits good crystallinity, matching the simulated pattern, whereas the melt-quenched *G*-Mg-adp only showed diffuse scattering peaks. Also, in-situ VT-XRPD data reveals that *C*-Mg-adp transition into dried structure begins to lose crystallinity around 225 °C and becomes completely amorphous after exceeding 265 °C.

Upon reaching its $T_m$, *C*-Mg-adp was melted and transformed into the glassy monolithic *G*-Mg-adp through rapid quenching in argon atmosphere (Fig. 2c, d). Scanning electron microscopy (SEM) analysis showed that after vitrification through cooling to room temperature, shards of *G*-Mg-adp display a smooth surface (Fig. 2e). These observations directly support the occurrence of thermal melting in *C*-Mg-adp, as indicated by the DSC data. Furthermore, the presence of spike-like and puffed shapes observed during annealing above $T_g$ confirms the transition of Mg-adp into a viscous liquid state (Supplementary Figs. 5 and 6)[32].

To confirm that *G*-Mg-adp had a composition identical to that of *C*-Mg-adp, nuclear magnetic resonance spectroscopy (NMR) and infrared (IR) spectroscopy were conducted (Supplementary Figs. 7 and 8). The NMR spectrum indicated the presence of adipates after vitrification, and the IR spectrum confirmed that the carboxylate coordination bonds with metal ions were retained with redshift on $v(COO^-)$ vibration modes of *G*-Mg-adp. Thermogravimetric-gas chromatography-mass spectrometry (TG-GC-MS) analysis revealed that molten Mg-adp produces CO and cyclopentanone gases as partial decomposition products when exposed to its $T_m$ for 3 h. Nevertheless, glass foams obtained after 3 h exhibited the IR peak that was completely consistent with *G*-Mg-adp, suggesting the persistence of Mg−adipate bonds despite partial decomposition (Supplementary Fig. 9).

The atomic connectivity and structural correlation in *C*-Mg-adp and *G*-Mg-adp were probed using X-ray total scattering data ($I(Q)$) and PDFs ($G(r)$) (Supplementary Figs. 10−12 and Fig. 3)[12,33]. $I(Q)$ shows sharp Bragg peaks for *C*-Mg-adp, unlike *G*-Mg-adp, indicating the loss of the highly crystalline structure at *G*-Mg-adp. However, $G(r)$ revealed that the local coordination environments ($r < 5$ Å) of *G*-Mg-adp were nearly identical to those of *C*-Mg-adp. This corresponds to the short-range bonds and correlations between the ligand and Mg ions. As shown in Fig. 3, peaks 1 and 2 in $G(r)$ correspond to the C−O and C−C bond distances in one adipate ligand, and peak 3 corresponds to the Mg−O coordination bond. Notably, after melt-quenching process, a subtle increase in the bond distance is observed at the position of maximum intensity position for Mg−O (Peak 3), while the C−O distance (Peak 1) shortens at its peak. These results align with the occurrence of redshift in $v(COO^-)$ IR peaks after melt-quenching, indicating a weakening Mg−O bond strength[34]. Peak 4 corresponds well with the unconnected C···C distance and exhibit a slightly increase in *G*-Mg-adp, suggesting an expansion of the $^-O_2C−CH_2−CH_2$ bond angle in the adipate ligand

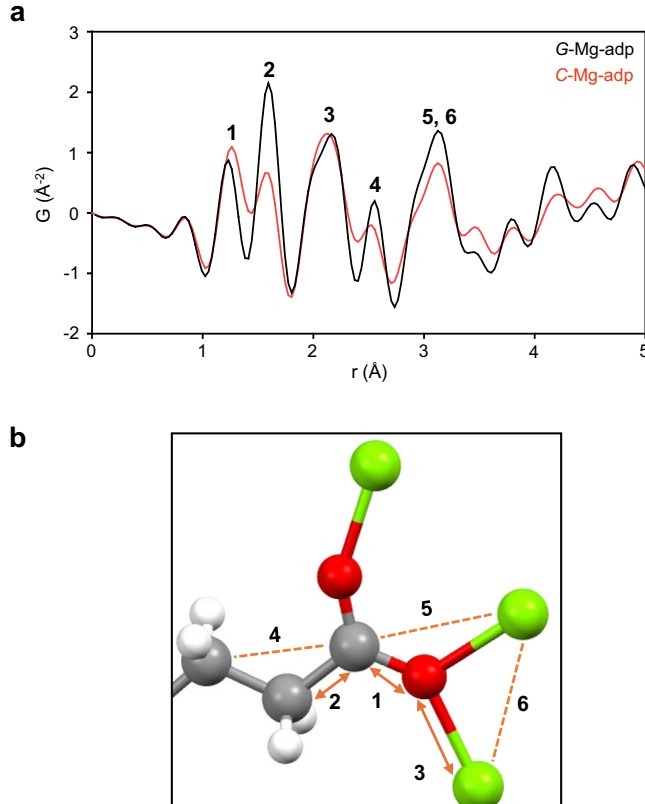

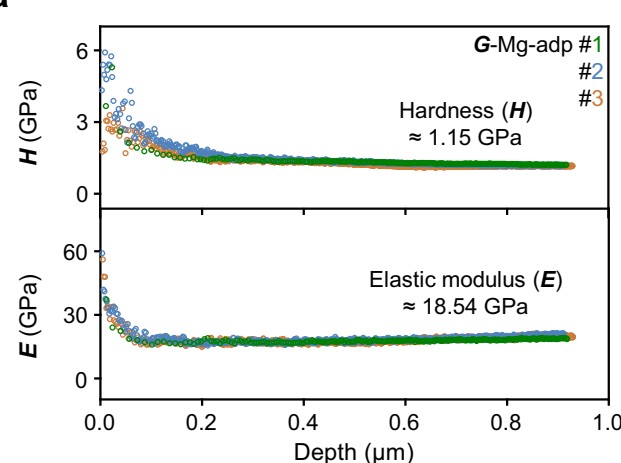

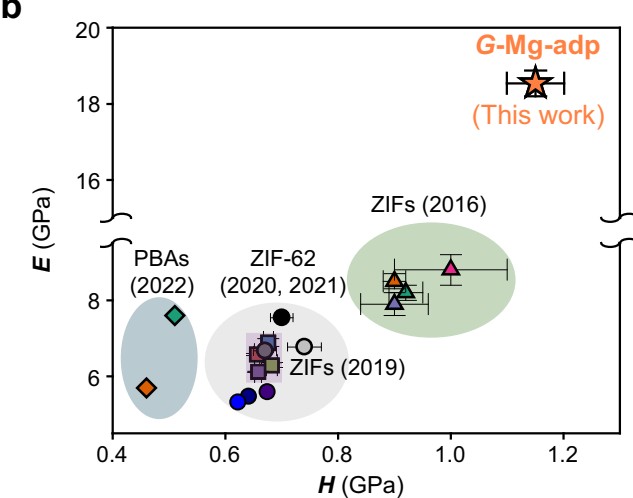

**Fig. 3 | Local structure analysis of Mg-adp with PDFs. a** The pair distribution function (PDF), $G(r)$, as a function of distance between atom pairs, denoted as $r$, for samples of $C$-Mg-adp (red) and $G$-Mg-adp (black). **b** Local coordination environments of Mg-adp. To represent various bonding modes of adipates with similar correlation distances, the half fragment of one adipate with $\mu^3 \cdot \eta^1 : \eta^2$ mode is shown. 1–6 in the figure represent the bonds and correlations between two atoms, respectively. Color scheme: C, grey; O, red; H, white; and Mg, light green.

during melting. Peak 5 and 6 are associated with C⋯Mg and Mg⋯Mg correlation distances, respectively. In a range of 4 – 5 Å, it is challenging to accurately attribute each peak to its specific origin because there are numerous correlations, but the main contributors are likely to be C⋯C, C⋯O, and O⋯O correlations (Supplementary Fig. 11)[35]. Overall, there is a slight increase in the Mg−O distance during the transformation from $C$-Mg-adp to $G$-Mg-adp, but the atomic correlation in local structures remains nearly the same for both. That is, the Mg-adipate connection maintains intact even after melt-quenching in $C$-Mg-adp.

**Mechanical properties of $G$-Mg-adp**

Understanding the mechanical properties of glassy materials is critical for designing and engineering glass-based products and providing insight into their structure-property relationships[36]. This has been assessed almost exclusively in ZIF glasses among coordination polymer glasses. Thus, we conducted a nanoindentation test on $G$-Mg-adp to understand the MOF glasses. This process involves pressing a small indenter into the surface of a sample and measuring the force and displacement during indentation. Using this, the load–depth curves of $G$-Mg-adp (Supplementary Fig. 13) were determined, resulting in the $H$ and $E$, as depicted in Fig. 4a. $G$-Mg-adp shows values of $H \approx 1.18 \pm 0.051$ and $E \approx 18.29 \pm 0.342$ GPa, recording the highest values above the reported coordination polymer glasses (Supplementary Table 2 and Fig. 4b)[16,37]. Notably, while the hardness improves in proportion to $T_g$ in conventional vitreous materials, $G$-Mg-adp exhibits higher hardness than ZIFs, even though its $T_g$ is less than 50 °C compared to that of ZIFs

**Fig. 4 | Mechanical properties of $G$-Mg-adp by nanoindentation tests.**
**a** Hardness ($H$) and elastic modulus ($E$) of $G$-Mg-adp as a function of nanoindentation depth under 20 mN maximum load with three tests. **b** Hardness-Modulus correlation of coordination polymer glasses. All data in the chart were evaluated using nanoindentation testing with the Oliver–Pharr method, except for the hardness of PBAs, which is measured using the Vickers hardness; error bars represent the reported standard deviations for each material. More details can be found in Supplementary Fig. 15 and Supplementary Table 1.

(Table 1 and Supplementary Fig. 14). This result can be interpreted, with caution, as being caused by both the highly deformed network structure in $G$-Mg-adp (see the discussion section below) and the stronger coordination bond of $G$-Mg-adp compared to ZIF glasses[38].

To compare the coordination bond strength in Mg-adp and ZIF-4, as a representative of meltable ZIFs, we conducted calculations by employing the ReaxFF reactive force field. The cluster models as well as crystal structure models for Mg-adp and ZIF-4 were constructed (Supplementary Fig. 16), from which the ligand was intentionally removed. In this system, the stabilization energy was calculated with the equation, $\Delta E = E_{tot} - (E_{MOF+} + E_{L-})$ ($E_{tot}$ = total system energy; $E_{MOF+}$ = energy of a charge MOF system; $E_{L-}$ = energy of a charged ligand), in which $\Delta E$ reflects the strength of metal−N or O coordinative bonds (Supplementary Table 3). For both models, $\Delta E$ of $C$-Mg-adp are larger negative values than those of ZIF-4, which means that coordination bonds in $C$-Mg-adp are stronger: an $\Delta E$ difference is approximately 4 and 57 kcal mol$^{-1}$, in the cluster and crystal model, respectively. The crystal model was further optimized to investigate mechanical properties, specifically the bulk modulus related to the glass transition

**Table 1 | Hardness, modulus, and their derivatives for MOF glasses and the organic crystal**

| Samples | G-Mg-adp | ZIF-4 | TIF-4 | ZIF-62 | ZIF-76 | Adipic acid [a] |
|---|---|---|---|---|---|---|
| $H$ (GPa) | 1.18 (± 0.051) | 0.92 (± 0.03) | 0.90 (± 0.06) | 0.656 (± 0.005) | 0.682 (± 0.01) | 0.3 |
| $E$ (GPa) | 18.29 (± 0.342) | 8.2 (± 0.2) | 7.9 (± 0.3) | 6.58 (± 0.02) | 6.29 (± 0.07) | 10.39 |
| $H/E$ (arb. u.) | 0.065 | 0.112 | 0.114 | 0.097 | 0.108 | 0.029 |
| $H^2/E$ (GPa) | 0.076 | 0.103 | 0.102 | 0.065 | 0.074 | 0.0087 |
| $T_g$ (°C) | 242 | 292 | 343 | 318 | 310 | - |
| Reference | This work | 12 | 12 | 38 | 38 | 47 |

[a] Values corresponding to the (110) face of adipic acid.

temperature. Density functional theory (DFT) calculations were performed using the CASTEP package[39,40]. The results revealed that ZIF-4 exhibited a comparatively larger bulk modulus of 29.9 GPa, surpassing that of C-Mg-adp which shows a bulk modulus of 13.2 GPa. Our calculation shows similar results with previously reported classical simulations utilizing the ReaxFF reactive force field yielded similar results; however, only the bulk modulus defined by Reuss showed a trend consistent with DFT calculations (Supplementary Table 4)[41]. It is worth noting that all measures of bulk modulus for defect crystal structures at 300 K, which is the simulated structures of the glass MOF, consistently produced similar outcomes. Interestingly, Mg-adp with a defect displayed a negative value for the bulk modulus, indicating an easy glass transition and showing good agreement with the experimental results. The results also agree well with the higher $\Delta H_{fus}$ and $E$ values of G-Mg-adp[42–44].

Higher hardness implies that the material is stronger; however, it can be easily fractured. Meanwhile, a higher $H^2/E$ ratio indicates greater external stress tolerance until fracture[45]. As shown in Table 1, G-Mg-adp demonstrated similar or higher $H^2/E$ values than those in previous studies, resulting in its enhanced strength and toughness over existing coordination polymer glasses. Compared to its originated organic ligand crystal of adipic acid, $E$ and $H$ of G-Mg-adp increased by about 1.6 and 3.7 times more than $E(110)$ and $H(110)$ of adipic acid crystal[46], suggesting that MOF glass can contribute to advancing the yield strain.

## Thermal Behavior of M-adp (M=Mn²⁺, Co²⁺, Tb³⁺)

The strong metal-carboxylate bond strength on C-Mg-adp is responsible for its high $\Delta H_{fus}$ (134.5 kJ/mol), which is attributed to the strong coordination between metal ions and carboxylate ligands but the lack of CFSE character of magnesium cation may relieve the enthalpy gap between the framework and its dissociation form due to kinetically labile bonds[47,48]. To better understand the effect of CFSE and metal-ligand bond strength on MOF melting, we studied the thermal behavior of series of C-M-adp (M = Mn²⁺, Co²⁺, and Tb³⁺)[49] upon heating (Fig. 5 and Supplementary Figs. 17–19). As shown in Fig. 5, only amorphization and decomposition were observed in C-Co-adp (Fig. 5e, f) and C-Tb-adp (Fig. 5h, i), while C-Mn-adp melted and transformed into G-Mn-adp during the cooling process (Fig. 5b, c). The DSC data indicated that C-Mn-adp exhibited $T_m$ of 238 °C and $T_g$ of 179 °C (Supplementary Fig. 20) Also, in-situ VT-XRPD of C-Mn-adp indicated the absence of the crystalline peak after 245 °C, showing a tendency consistent with the DSC results (Supplementary Fig. 21). Mn K-edge X-ray absorption fine structure (XAFS) measurements revealed that G-Mn-adp exhibits a structure with shorter Mn-Mn distance than C-Mn-adp, while maintain Mn-O and Mn-Mn distance within local structure of C-Mn-adp (Supplementary Fig. 22). The shorter Mn-Mn distance and TG-GC-MS data suggest partial decomposition in molten Mn-adp, while NMR data confirm that the adipate ligand remains in

G-Mn-adp, even when Mn-adp exposed to $T_m$ for 3 hours (Supplementary Figs. 23 and 24). These results imply that G-Mn-adp contains some unknown decomposition products that were not detected in the XRPD data.

The non-melting behavior of C-Co-adp and the low $T_m$ of C-Mn-adp, as compared to C-Mg-adp, could partially be attributed to the CFSE[47] and ion radius of the metal ions[50]. Moreover, the absence of melting in C-Tb-adp, which has a high oxidation number metal node, may be attributed to its high metal-ligand dissociation energy[51]. Besides the type of metal, various factors can influence the $T_m$ of a MOF. The C-M-adp series features a similar 1D infinite M-O chain as the SBU; however, they do not have an isoreticular structure as carboxylate can have varying binding modes. Specifically, C-M-adp series, including Mg, mostly exhibits a 3D structure, while C-Mn-adp forms a 2D structure where the interconnection of SBUs extends only in the form of a sheet (Fig. 5a, d, and g). Since structural influences can be significant in the thermodynamic behavior of MOFs, a direct comparison of thermal trends between these series may be somewhat inaccurate. Nevertheless, the melting behavior of C-Mg-adp and C-Mn-adp can be compared to the tendency that most of the stable liquid CP/MOFs discovered so far contain d¹⁰ metal, suggesting that s-block and d⁵ metals can be candidates for meltable framework design[15].

More important driving force for the low $T_m$ of C-Mg-adp and C-Mn-adp is the larger entropic benefit resulting from the aliphatic moiety[13,52]. Aliphatic carboxylate ligands have many configurations in the liquid phase of the framework than aromatic carboxylates owing to their low symmetry value[53–56]. Additionally, the rotationally flexible alkyl chain moiety allows the transformation of the porous framework during heating, resulting in the formation of pore-collapsed structure that can reduce the residual ligand entropy in the solid phase (Supplementary Figs. 4 and 27)[14,44,57].

## Discussion

The glass-forming ability (GFA) of C-Mg-adp ($T_g/T_m = 0.93$) is much higher than that of most members of the ZIF family as well as the empirical prediction of the Kauzmann "2/3" law[58], despite the fact that structurally comparable aliphatic amide-based networks have low GFAs, leading to recrystallization during cooling (Supplementary Fig. 28)[14,58]. This feature is due, in part, to the relatively strong coordinate bonds, which partially contribute to the stabilization of local structure in the molten phase of MOFs, resulting in a less fragile liquid[59–62]. In addition, the substantial flexibility of the aliphatic ligands in Mg-adp enables a high degree of deformation in C-Mg-adp structure, leading to the formation of a nonporous and dense phase upon the melting process. Since modulus and hardness are inversely proportional to porosity[63,64], this structural deformation is one of the factors contributing to the harder mechanical properties of G-Mg-adp compared to ZIF glasses, which have ultramicroporosity after vitrification[65]. Moreover, the coordination polymer Mg-adp, constructed through the infinite connection between the aliphatic chain and the 1D chain SBU is reminiscent of an organic polymer. The melt-quenching process could enhance the mechanical properties of G-Mg-adp by inducing structure entanglement and high packing density, akin to what is observed in semicrystalline polymers[66,67].

Furthermore, we observed that G-Mg-adp exhibited much higher water stability than C-Mg-adp (Supplementary Fig. 30). While C-Mg-adp rapidly degrades in a small amount of water, G-Mg-adp almost retains its shape and remains immersed even with an excess quantity of water. As revealed in the IR spectra, after soaking in water, the filtered G-Mg-adp (G-Mg-adp-h) maintains the coordination bond between Mg²⁺ and carboxylate. A red shift was observed in the $v(COO^-)$ modes of G-Mg-adp-h, but they revert to the original state after activation at 80 °C in a vacuum. Throughout the entire process, G-Mg-adp consistently maintains an amorphous states and does not undergo

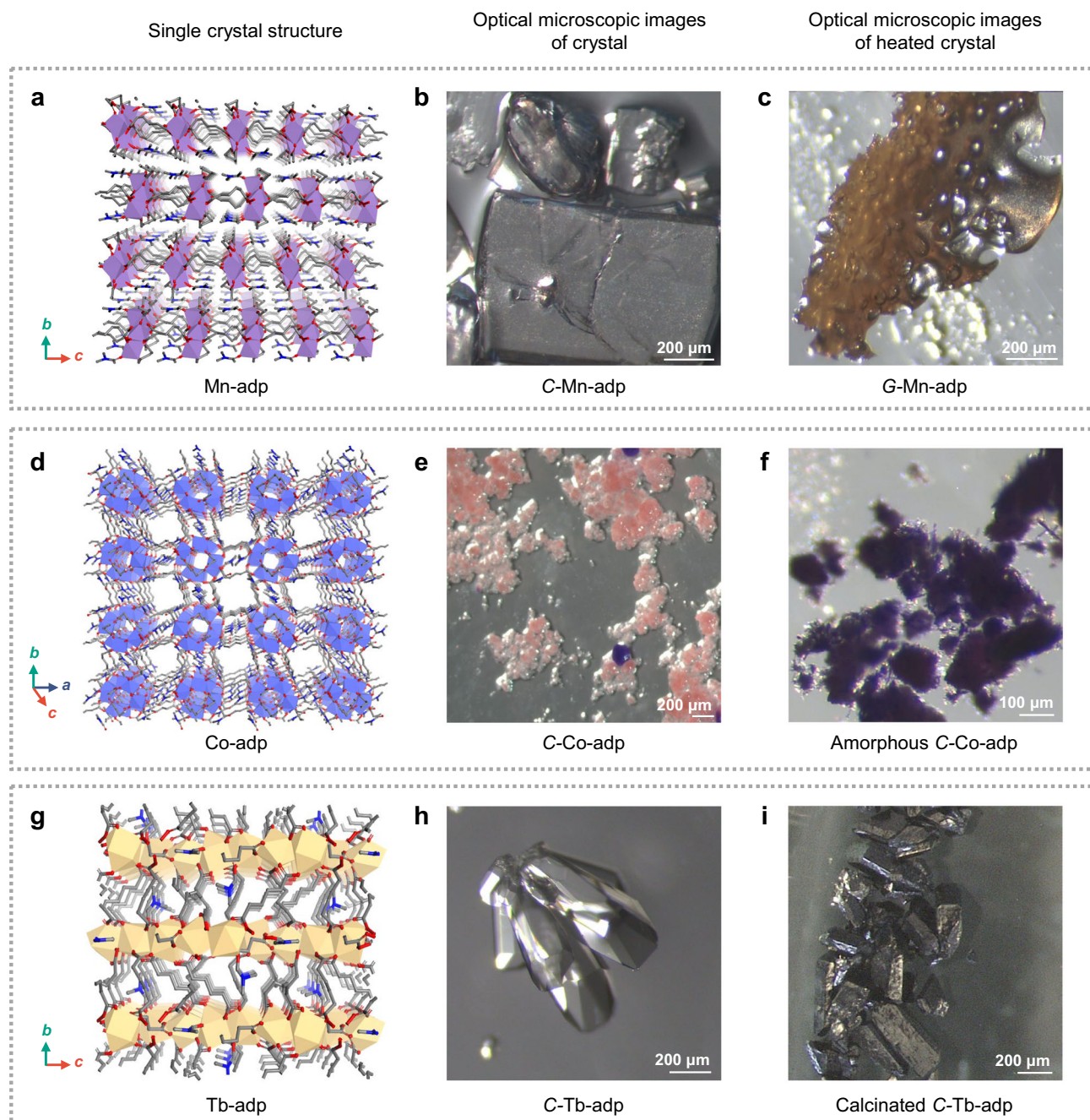

**Fig. 5 | Variations in the thermal transition of M-adp (M=Mn²⁺, Co²⁺, Tb³⁺).** Single-crystal structures of M-adp (**a**, **d**, and **g**), optical microscope images of *C*-M-adp (**b**, **e**, and **h**), and optical microscope images of *C*-M-adp subsequent to thermal conversion in an inert gas (**c**, **f**, and **i**). *C*-Mn-adp could form *G*-Mn-adp through a melt-quenching process (**c**), while *C*-Co-adp undergoes amorphization by desolvation (**f**), and *C*-Tb-adp were maintained the morphology of the crystal until calcination (**i**). Color scheme: C, grey; O, red; N, blue; Mn, purple; Co, light blue; and Tb, beige.

crystallization. The origin of the improved water stability in MOF through vitrification may be revealed through detailed structural analysis of *G*-Mg-adp, *G*-Mg-adp-*h*, and activated *G*-Mg-adp-*h*. However, a complete survey of these topics is beyond the scope of this paper. Nonetheless, since research on the distinctions in properties between crystals and glasses of CPs/MOFs is still in its infancy, further studies are needed to elucidate the structural transformations occurring during the melting process that influence the chemical stability of glasses.

In summary, we report the discovery of the carboxylate-based MOF glasses obtained via the melt-quenching of a crystalline 3D MOFs. The melting of *C*-Mg-adp and *C*-Mn-adp can likely be attributed the high entropy contribution of the aliphatic ligand and the low CFSE of the magnesium and manganese ions. Furthermore, we demonstrate that *G*-Mg-adp exhibits unique mechanical properties compared to ZIF glasses, owing to the relatively strong coordination bonds of the carboxylate group and its nonporous nature. These results provide valuable insights into the structure-property relationship of MOF glasses. Our study not only expands the range of liquid/glass MOF materials but also provides a promising approach for the development of meltable MOF structures based on carboxylate linkers, which are widely present in MOFs.

## Methods

### Thermal vitrification of *C*-M-adp

*C*-M-adp was hand-grinded before heating, to make bulk powder. The powder samples were placed into a crucible or on a slide glass. The prepared samples were put into a tube furnace and then heated at 10 °C min$^{-1}$ under argon flow of 100 mL min$^{-1}$. After reaching the target temperature of $T_m$ or $T_d$, the heating was stopped, and the samples were cooled down naturally under inert gas flow to room temperature.

### Mechanical properties measurement

All measurement samples were prepared by placing *C*-Mg-adp powder between glass slides, clamping the slides with forceps, and then subjecting them to a 10-minute melting process under inert gas conditions. After quenching, the glass slides were separated. All hardness and elastic modulus data were measured by a nanoindenter from Anton-Paar with a Berkovich tip based on the Oliver-Pharr method. Maximum force (load) was 20 mN and a sinusoidal method (known as a continuous stiffness measurement method) was applied in order to measure the variations of hardness and elastic modulus with indentation depth. Since hardness and elastic modulus decreased in low indentation depth due to indentation size effect, their saturated values were selected as the representative values in Supplementary Table 2.

## Data availability

The data that support the findings of this work are presented in the Letter and the Supplementary Information. Additional data are available on request from the corresponding authors. Source data are provided with this paper.

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

## Acknowledgements

This work was supported by the National Research Foundation of Korea (NRF) grant (NRF-2020R1A2C3008908 and NRF-2019M3E6A1103980 to H.R.M.) and Korea Basic Science Institute (National Research Facilities and Equipment Center) grant funded by the Ministry of Education (2020R 1A 6 C 101B194 to H.R.M.).

## Author contributions

M.K. and H.R.M. conceived the idea for the project. M.K. designed the experiments and performed the synthesis and characterization of the crystalline and glass MOFs. H.-S.L. and D.-H.S. analyzed the mechanical properties of the glass MOF under direction of E.-c.J.; S.J.C. simulated models for MOFs and calculated the coordination bond strength and mechanical properties of MOFs. M.K. and H.R.M. wrote the paper, and all the authors contributed to preparing the manuscript.

## Competing interests

The authors declare no competing interests.
