## [Peer Review File · Nature Communications]

Melt-quenched Carboxylate Metal–Organic Framework GlassesReviewers' Comments:

Reviewer #1:

Remarks to the Author:

This study reports the melting and vitrification of MOFs constructed from dicarboxylate ligands. Recent papers (10.1021/jacs.2c04532, 10.26434/chemrxiv-2023-05kv3) have reported that MOFs composed of ligands with at least one carboxylate group transform to a glassy state, but they are aromatic carboxylates and do not show a liquid phase upon heating. The present work shows melting due to the aliphatic ligand structure. The compounds are based on previous reports by the authors and the crystal structure, thermal behaviour and thermodynamic considerations are well discussed. In conclusion, it is encouraging that there may be more design principles for glass and liquid formations from carboxylate MOF crystals. On the other hand, I have some concerns and suggestions, which are listed below. Particularly with regard to thermal stability, which is related to the compositions and structures of the glasses.

(1) The authors used this Mg-MOF to prepare hierarchically nanoporous MgO by thermal conversion (J. Am. Chem. Soc. 2013, 135, 24, 8940). Both Fig. S5 of the published work and Fig. S4 of the present work show that the Mg-MOF undergoes structural changes during desolvation. As a precursor to crystal melting, it is important to understand the true structure of the guest-free state. In this work, the XRD pattern of the as-synthesised sample and the simulated patterns do not match well. For example, there is a clear unknown peak between 8.9-10 degrees, and the authors need to further confirm the presence of impure phases in the sample, which would have a non-negligible influence on the melting behaviour.

(2) The colour of the vitrification products, especially G-Mn-adp, is much darker compared to the original crystal, suggesting partial decomposition. To investigate this, I suggest characterisation by long-term TG at close to the T_m , TG-MS to detect CO₂ or other gases from ligands, ¹H NMR of the sample after acid digestion, variable temperature IR to monitor carboxylate behaviour.

3) Glass structure characterisation is mainly based on PDF analysis. The local coordination environments ($r < 6 \text{ \AA}$) are not consistent with the crystal structure (Fig. 2). The correlation between peaks 1,2,3 and the structure is confusing. I think XAS measurements for glassy and molten states as well as crystals are essential.

4) The authors conclude that "The highest hardness of the glass measured in the study among the other reported MOF glasses can be interpreted due to the stronger coordination bonds in G-Mg-adp than in ZIF glasses". This is misleading, although this is one of the key discussions in this paper. We know that it is difficult to quantify the actual coordination bond strength in MOFs. Thermal and chemical stabilities depend strongly on the network topology, density and so on. I recommend doing any experiments or calculations to support this speculation, such as DFT to model the bond strength based on crystal structures, VT spectroscopy (Raman?) to observe lattice vibrations. I also note that the preparation procedure and the resulting monoliths vary the hardness values. More details on the preparation should be provided.

(5) They have synthesised the C-M-adp (M = Mn²⁺, Co²⁺ and Tb³⁺) using the same organic ligand, but have not provided a structural description. Figure S15 shows that the structures of C-M-adp are different from those of C-Mg-adp. It is not fair to compare the properties using different structures. Is the structure of G-Mn-adp the same as that of G-Mg-adp? Further structural investigations may help to explain the different mechanical properties.

Some technical notes.

- For H₂ and CO₂ sorption, the x-axis is the relative pressure (P/P₀) and it is unclear what their P₀ values are.

- The SEM images in Fig. S5 show that the surface of G-Mg-adp becomes very uneven after heating for 1 h. Any reason?
- What is the stability of the glasses in water/humidity?
- TGA of pristine C-Mg ADP shows that Td was evaluated at 330°C (Fig. S1), while TGA of the obtained glass showed that Td was around 280°C (Fig. S2). Can you explain these differences? I think that the marked position of Td in Fig. S1 is inappropriate because a significant weight change occurs. Td should be around 280 °C and it is close to the marked Tm (283 °C).

Reviewer #2:

Remarks to the Author:

In this work, the authors investigated the thermal behaviors of a family of rod-spacer type MOFs, which is constructed from a metal carboxylate chain linked by an aliphatic carboxylate ligand of $-(\text{OOC})(\text{CH}_2)_4(\text{COO})-$. With the aim of finding new MOF glasses, the Mg and Mn based MOF seem to be melted and can produce MOF glass upon quenching. The melting and glass forming behaviors are mainly supported by TGA, DSC, PXRD, SEM and other characterization results. In contrast, direct decomposition or amorphization were observed in iso-reticular MOFs with Co and Tb. In deeded, liquid and glass states represent the great advance and interesting new direction of MOFs, considering the different aspects of MOF processing, new MOF function and understanding the nature of glass. Hence, the presented results are valuable for this area and can be considered for publication. However, from the viewpoint of novelty, these findings of melt-quenched carboxylate metal-organic framework glasses are not surprising or beyond expectation. Both melting-quenching and other strategies like mechanical, perturbation methods are proved to be useful to produce glasses, form different types of MOF precursor. In addition, there are several experimental queries on the real glass formation and glass structure. On this occasion, the current manuscript is still too preliminary and major revision is necessary to support it's publication.

1. Comparing to the inserted 2nd heating DSC which seems normal, the quite flat DSC curve for the first heating run is strange. Have the authors performed some data treatment for the raw signals?
2. I noticed a heating rate of 10 K min⁻¹ was used for all the DSC measurements. However, In Figure S1 and S2, the TGA used a much lower heating rate of 1 K min⁻¹. Why the same heating rate are not used. The different heating rate will affect the key parameters of Tm, Tg and Td, and leads to data mismatching?
3. Usually, the glass transition or the DSC curve is repeatable upon heating. However, there is just one DSC curve for the quenched same. It is difficult to estimate the stability of the glass state?
4. In figure 1c, there is an evident peak at 2theta of 7.5o for G-Mg-adp. How about the measured intensity? If it is a strong diffraction peak, it is somewhat contradictory to a glass state.
5. In figure S3, the glass transition point can not be well determined in the DSC curves with high heating rate above 30 K min⁻¹. Then, the accuracy of fragility is not high.
6. The in-suit VT-XRPD data of C-Mg-adp is quite useful and significant evidence to deduce the structure transformation process upon heating. However, the presented data in Figure S4 is too coarse. Especially, if the DSC endothermic peak between 283 and 300 oC is stemming from melting, the PXRD at that temperature range should have no peaks for a liquid state.
7. Another great query is about the local structure analysis of Mg-adp with PDFs. Through the authors give long sentence discussion, for current result, only the short-range atom correlations below 5 angstroms give valuable information. It seems the data are X-ray total scattering data are not well collected, leading to low data resolution at middle or long range. On this case, the analysis for the peak of b, c, d, e, f is meaningless, because we can not definitely say there is or is not a peak.
8. Comparing to the Mg-MOF, the characterization for Mn-adp is far from sufficient. We can not deduce what kind of structure transformation occurs, based on presented limited result. In figure S17, the Tm = 238 Oc is given for C-Mn-adp, how about the standard to estimate melting in the continuous DSC curves?
9. The relevance between liquid/glass forming and chemical/structure character is essential and important for MOF glass research. In this work, the Mg and Mn MOF behave in different fashion,

comparing to the Co and Tb compound. However, the presented explanation is too general without data supports.

10. The authors have pointed out the low fragility index of 26.3 implies that the flowing is hardly observed for the MOF glass. But there is a stalactite-like and puffed shapes were found in Figure S16 for G-Mg-adp. How such a image was obtained?

11. On the whole, the data quality, organization and presentation are poor, additional experiment and careful manuscript revision are absolutely necessary. For example, in figure S5, "SEM images of G-Mg-adp was heated at T_m for 0 minutes (top) and 1 hour (bottom)", what is the meaning for these data? Similar, in Figure S19, 0 minutes? In Figure S15, what is the exact treatment temperature of each samples?

Reviewer #3:

Remarks to the Author:

The presented studies provide a very first example of a carboxylate-based MOF glass. In general, the reviewer recommends the publication of this paper after addressing certain issues that have been discussed below. The reviewer acknowledges and agrees with the authors' statement that this is the first example of its kind. However, it remains unclear why this particular system had not been prepared earlier, considering that carboxylate-containing MOFs are one of the most extensively investigated classes. While the reviewer is not debating that it is a first example, it is important to emphasize in the introduction why these studies could be useful.

The reviewer also suggests including the table of known the imidazole-containing MOF glasses with the relevant porosity and surface area. The reviewer commends the authors on their thorough analysis, but recommends comparing the findings, whenever possible, with existing ZIF-based glass systems. While some tables are already provided and prove to be highly useful, they alone may not be sufficient in certain cases.

The Reviewer recommends improving the quality of the illustrative content in the paper. The figures appear to have a relatively low quality, with instances of font mismatching between the figures and the main text, which hinders the clarity of certain labels.

Could the authors re-check the fit (PDF analysis) presented in Figure 2?

It is not clear to the reviewer what is intended to be observed based solely on the presented figures? What parts of the presented pictures are important? The reviewer suggests modifying the main text and the corresponding figure caption.

Response to Reviewers

Reviewer #1

This study reports the melting and vitrification of MOFs constructed from dicarboxylate ligands. Recent papers (10.1021/jacs.2c04532, 10.26434/chemrxiv-2023-05kv3) have reported that MOFs composed of ligands with at least one carboxylate group transform to a glassy state, but they are aromatic carboxylates and do not show a liquid phase upon heating. The present work shows melting due to the aliphatic ligand structure. The compounds are based on previous reports by the authors and the crystal structure, thermal behaviour and thermodynamic considerations are well discussed. In conclusion, it is encouraging that there may be more design principles for glass and liquid formations from carboxylate MOF crystals. On the other hand, I have some concerns and suggestions, which are listed below. Particularly with regard to thermal stability, which is related to the compositions and structures of the glasses.

We appreciate the reviewer's positive recommendation and constructive comments, and we have carefully addressed all of them as below.

(1) The authors used this Mg-MOF to prepare hierarchically nanoporous MgO by thermal conversion (J. Am. Chem. Soc. 2013, 135, 24, 8940). Both Fig. S5 of the published work and Fig. S4 of the present work show that the Mg-MOF undergoes structural changes during desolvation. As a precursor to crystal melting, it is important to understand the true structure of the guest-free state. In this work, the XRD pattern of the as-synthesised sample and the simulated patterns do not match well. For example, there is a clear unknown peak between 8.9-10 degrees, and the authors need to further confirm the presence of impure phases in the sample, which would have a non-negligible influence on the melting behaviour.

(Response) We appreciate the reviewer's comments. *C*-Mg-adp is a highly flexible MOF, which shows significant structural transformations depending on the amounts of guest molecules and temperatures, as evidenced by VT-XRD patterns (Review Only Figure R1). When we carefully prepared the sample for VT-XRD measurements to prevent the loss of guest molecules, the small peak between 8.9-10° was scarcely observed in the XRD pattern obtained at 25 °C. However, as the temperature increased, the peak gradually increased at 45 °C and shifted higher at 65 °C, ultimately vanishing at 85 °C. This suggests that the peak might be attributed to the flexibility of *C*-Mg-adp. Regrettably, we couldn't entirely eliminate the appearance of this peak in the as-synthesized *C*-Mg-adp due to the loss of guest molecules during grinding. Nonetheless, we are confident that this peak does not impact the overall glass transition of *C*-Mg-adp.

Review Only Figure R1. VT-XRPD patterns of C-Mg-adp indicated as relative intensity. The data were collected at intervals of 20 °C in the range of 25 °C to 345 °C.

(2) The colour of the vitrification products, especially G-Mn-adp, is much darker compared to the original crystal, suggesting partial decomposition. To investigate this, I suggest characterisation by long-term TG at close to the T_m , TG-MS to detect CO₂ or other gases from ligands, ¹H NMR of the sample after acid digestion, variable temperature IR to monitor carboxylate behaviour.

(Response) We appreciate the reviewer's comments and suggestions. In order to examine the structural/chemical stability of Mg-adp and Mn-adp, long-term TG-GC-MS and FT-IR spectra were obtained upon heating at temperatures close to the T_m of each MOF (285 °C for Mg-adp and 240 °C for Mn-adp, respectively) for 3 h. As shown in Supplementary Fig. 9 and 22, both MOFs show slight weight loss in the isothermal region at T_m , which corresponds to CO, CO₂, DMA, and other organic species. Even after heat treatment, both MOFs retain the coordination bonds between the metal centers and carboxylate groups of the adp ligands, as evidenced by the FT-IR spectra. In addition, ¹H NMR spectrum for Mn-adp after the heat treatment and acid digestion shows that no other decomposed species are present in the solid (Supplementary Fig. 23). This implies that the MOF glasses are connected by metal-ligand coordination bonds, even though partial decomposition can be observed. This may be the reason of the color change in G-Mn-adp, in which a trace of Mn⁴⁺ species can be formed during the heat treatment.

- Change made: TG-GC-MS and FT-IR spectra for Mg-adp and Mn-adp were added in Supplementary Fig. 9 and Supplementary Fig. 22, respectively. ^1H NMR spectra for a series of Mn-adp were given in Supplementary Fig. 23. Also, the corresponding statements were added in the text.

Supplementary Fig. 9 a, TG-GC-MS data were collected for C-Mg-adp by heating up to 285 °C, followed by isothermal measurements for 3 h. The types of species detected for each range were represented at the top. **b**, The FT-IR data collected after the heat treatment of Mg-adp, compared with C-Mg-adp and G-Mg-adp.

Supplementary Fig. 22 a, TG-GC-MS data were collected for *C-Mn-adp* by heating up to 240 °C, followed by isothermal measurements for 3 h. The types of species detected for each range were represented at the top. **b**, The FT-IR data collected after the heat treatment, compared with *C-Mn-adp* and *G-Mn-adp*.

Supplementary Fig. 23 ^1H NMR spectra for *C-Mn-adp*, *G-Mn-adp* and the sample after heat treatment at 285 °C for 3 h.

3) Glass structure characterisation is mainly based on PDF analysis. The local coordination environments ($r < 6 \text{ \AA}$) are not consistent with the crystal structure (Fig. 2). The correlation between peaks 1,2,3 and the structure is confusing. I think XAS measurements for glassy and molten states as well as crystals are essential.

(Response) We appreciate the valuable feedback provided by the reviewer. We re-checked the PDF and more clearly marked the correlations for each PDF peak in the figure.

The modified PDF analysis considers only short-range orders ($r < 5 \text{ \AA}$) as reliable values, considering the Q_{max} value and resolution of lab-scale equipment. Also, the peak appeared around 0.9 \AA was the result of background noise, and so the previous assignment of this peak as C-H bonds has been deleted. In addition, we added explanation of the PDF results for the local structure of C-Mg-adp and G-Mg-adp. $G(r)$ revealed that the local coordination environments ($r < 5 \text{ \AA}$) of G-Mg-adp are nearly identical to those of C-Mg-adp. The differences in peaks attributed to Mg-O, C-O and C-C were explained in the main text. Especially, unconnected C-C distances can be varied upon melting due to the freely rotatable C-C single bonds in the aliphatic linker. This phenomenon is also observed in coordination polymers composed of aliphatic amide. (refer to Supplementary Fig. 39 in *J. Am. Chem. Soc.* **143**, 7, 2801–2811 (2021).)

We endeavored to confirm the local structure around magnesium in Mg-adp through XAS and solid ^{25}Mg NMR; unfortunately, these measurements were not possible due to limitations in the available equipment. The K-edge value of magnesium, a light metal, is at 1.3 keV , which falls into an energy range not well supported by general XAS equipment (*Am. Mineral.* **93**, 495-498 (2008). *Chem. Mater.* **22**, 161-166 (2010)). ^{25}Mg NMR is also challenging due to the unfavorable properties of ^{25}Mg , including a low gyromagnetic ratio, low natural abundance, and high quadrupolar moment (*Phys. Chem. Chem. Phys.* **19**, 613-625 (2017)).

For Mn-adp, where XAS measurement is possible, changes in the local structure near Mn after melting were identified through FT-XAFS data.

- Change made:

- Fig. 3 (Fig 2 in the previous version) was modified for clarity, and the explanation of PDFs for C-Mg-adp and G-Mg-adp was revised as below.

“As shown in **Fig. 3b**, peaks 1 and 2 in $G(r)$ correspond to the C–O and C–C bond distances in one adipate ligand, and peak 3 corresponds to the Mg–O coordination bond. Notably, after melt-quenching process, a subtle increase in the bond distance is observed at the position of maximum intensity position for Mg–O (peak 3), while the C–O distance (peak 1) shortens at its peak. These results align with the occurrence of redshift in $\nu(\text{COO}^-)$ IR peaks after melt-quenching, indicating a weakening Mg–O bond strength.³⁴ Peak 4 corresponds well with the unconnected C–C distance and exhibit a slightly increase in G-Mg-adp, suggesting an expansion of the $-\text{O}_2\text{C}-\text{CH}_2-\text{CH}_2$ bond angle in the adipate ligand during melting. Peak 5 and 6 are associated with C–Mg and Mg–Mg correlation distances, respectively. In a range of 4 – 5

Å, it is challenging to accurately attribute each peak to its specific origin because there are numerous correlations, but the main contributors are likely to be C···C, C···O, and O···O correlations (**Supplementary Fig. 11**).³⁵ Overall, there is a slight increase in the Mg-O distance during the transformation from C-Mg-adp to G-Mg-adp, but the atomic correlation in local structures remains nearly the same for both. That is, the Mg-adipate connection maintains intact even after melt-quenching in C-Mg-adp.”

Fig. 3 | Local structure analysis of Mg-adp with PDFs. **a**, $G(r)$ for samples of C-Mg-adp (red) and G-Mg-adp (black). **b**, Local coordination environments of Mg-adp. To represent various bonding modes of adipates with similar correlation distances, the half fragment of one adipate with $\mu^3-\eta^1:\eta^2$ mode is shown. 1-6 in the figures represent the bonds and correlations between two atoms, respectively. Color scheme: C, gray; O, red; H, white; and Mg, light green.

- We added the XAS results for Mn-adp to understand the structural changes upon melting.

“Mn K-edge X-ray absorption fine structure (XAFS) measurements revealed that G-Mn-adp exhibits a structure with shorter Mn-Mn distance than C-Mn-adp, while maintain Mn-O and Mn-Mn distance within local structure of C-Mn-adp (**Supplementary Fig. 21**).”

Supplementary Fig. 21 The wavelet contour using Morlet method with $\eta = 10$ and $\sigma = 1$ over XAFS data of **a**, C-Mn-adp and **b**, G-Mn-adp. **c**, The Fourier-transforms of XAFS spectra as a function of radial distance for C-Mn-adp (red) and G-Mn-adp (black). Phase shift within the data set was uncorrected.

4) The authors conclude that "The highest hardness of the glass measured in the study among the other reported MOF glasses can be interpreted due to the stronger coordination bonds in G-Mg-adp than in ZIF glasses". This is misleading, although this is one of the key discussions in this paper. We know that it is difficult to quantify the actual coordination bond strength in MOFs. Thermal and chemical stabilities depend strongly on the network topology, density and so on. I recommend doing any experiments or calculations to support this speculation, such as DFT to model the bond strength based on crystal structures, VT spectroscopy (Raman?) to observe lattice vibrations. I also note that the preparation procedure and the resulting monoliths vary the hardness values. More details on the preparation should be provided.

(Response) We appreciate the valuable feedback provided by the reviewer. According to the reviewer's comments, we conducted theoretical calculations to compare the coordination bond strength in C-Mg-adp and ZIF-4. The cluster and crystal models in which the ligand is intentionally removed from the original structures were constructed to calculate the stabilization energy, $\Delta E = E_{tot} - (E_{MOF^+} + E_{L^-})$, which reflects the bond strength of metal and ligands. The results clearly show that for both models, ΔE of C-Mg-adp are larger negative values than those of ZIF-4, which means that coordination bonds in C-Mg-adp are stronger. In addition, the theoretical calculation suggested that the bulk modulus related to the glass transition temperature. We added the calculation results and Figures as shown below. Also, we provided more details for hardness measurements.

- Change made:

- Added theoretical calculation results:

“To compare the coordination bond strength in Mg-adp and ZIF-4, as a representative of ZIFs, we conducted calculations by employing the ReaxFF reactive force field. The cluster models as well as crystal structure models for Mg-adp and ZIF-4 were constructed (**Supplementary Fig. 15**), from which the ligand was intentionally removed. In this system, the stabilization energy was calculated with the equation, $\Delta E = E_{tot} - (E_{MOF^+} + E_{L^-})$ (E_{tot} = total system energy; E_{MOF^+} = energy of a charge MOF system; E_{L^-} = energy of a charged ligand), in which ΔE reflects the strength of metal–N or O coordinative bonds (**Supplementary Table 3**). For both models, ΔE of C-Mg-adp are larger negative values than those of ZIF-4, which means that coordination bonds in C-Mg-adp are stronger: an ΔE difference is approximately 4 and 57 kcal mol⁻¹, in the cluster and crystal model, respectively. The crystal model was further optimized to investigate mechanical properties, specifically the bulk modulus related to the glass transition temperature. Density functional theory (DFT) calculations were performed using the CASTEP package.^{39,40} The results revealed that ZIF-4 exhibited a comparatively larger bulk modulus of 29.9 GPa, surpassing that of C-Mg-adp which shows a bulk modulus of 13.2 GPa. Our calculation shows similar results with previously reported classical simulations utilizing the ReaxFF reactive force field yielded similar results; however, only the bulk modulus defined by Reuss showed a trend consistent with DFT calculations (**Supplementary Table 4**).⁴¹ It is worth noting that all measures of bulk modulus for defect crystal structures at 300 K, which is the simulated structures of the glass MOF, consistently produced similar outcomes. Interestingly, Mg-adp with a defect displayed a negative value for the bulk modulus, indicating an easy glass transition and showing the good agreement with the experimental results.”

Supplementary Fig. 15 Model construction by removing a ligand from the original structures. **a**, A cluster model and **b**, a crystal model of ZIF-4. **c**, A cluster model and **d**, a crystal model of C-Mg-adp.

Supplementary Table. 3 Stabilization energy for metal-ligand formation of ZIF-4 and Mg-adp.

sample	ΔE (kcal mol ⁻¹) ^a	
	Cluster model	Crystal model
ZIF-4	-35.4	-65.3
Mg-adp	-39.0	-113.8

^a ΔE can be obtained following the equation: $\Delta E = E_{tot} - (E_{MOF^+} + E_{L^-})$

Supplementary Table. 4 Lattice parameters for the initial, optimized and defect crystal structures.

		a (Å)	b (Å)	c (Å)	α (°)	β (°)	γ (°)	Bulk Modulus (GPa)
ZIF-4	Initial	15.3073	18.4260	15.3950	90.00	90.00	90.00	-
	Optimized ^a	15.4423	18.4164	15.8302	90.00	90.00	90.00	29.9
	Optimized ^b	13.8823	17.8336	15.3667	90.00	90.00	90.00	0.86/1.44/1.15 ^c
	Defect ^d	13.9267	18.5674	15.1760	92.83	89.01	90.37	3.45/1.22/2.33 ^c
Mg-adp	Initial	13.4950	19.0163	25.9664	81.04	74.94	69.22	-
	Optimized ^a	13.4990	18.8970	26.0070	80.14	74.93	69.07	13.2
	Optimized ^b	14.2987	16.2385	26.3646	76.61	72.63	64.90	0.28/4.07/2.17 ^c
	Defect ^d	14.7082	16.8661	25.5742	74.02	71.49	64.15	- ^e

^aThe structure has been optimized readily using CASTEP program with OTF potential.

^bThe reactive force field has been utilized during the optimization employing *NPT* condition at 300 K with the General Utility Lattice Program.

^cThe bulk modulus (Reuss/Voigt/Hill conventions) contains information regarding the hardness of a material with respect to various types of deformation which are available from the GULP calculation.

^dThe structure has been optimized under the condition that charge in a periodic cell was neutralized with a background and the charge was obtained using iterative algorithms rather than matrix diagonalization.

^eThe defect structure is likely to be fragile because of the defect induced by the deletion of adp ligand and resulted in the negative bulk modulus.

- Added the measurement details for mechanical properties:

“All measurement samples were prepared by placing C-Mg-adp powder between glass slides, clamping the two glass slides with forceps, and then subjecting them to a 10-minute melting process under inert gas conditions. After quenching, the glass slides were separated.

(5) They have synthesised the C-M-adp (M = Mn²⁺, Co²⁺ and Tb³⁺) using the same organic ligand, but have not provided a structural description. Supplementary Fig. 15 shows that the structures of C-M-adp are different from those of C-Mg-adp. It is not fair to compare the properties using different structures. Is the structure of G-Mn-adp the same as that of G-Mg-adp? Further structural investigations may help to explain the different mechanical properties.

(Response) We appreciate the reviewer's comments and fully agree with the concern. We have reported many studies regarding the relationship between the structure of MOFs and their thermal behaviors (refer to 'Transformation of Metal-Organic Frameworks into Functional Nanostructured Materials: Experimental Approaches Based on Mechanistic Insight', *Acc. Chem. Res.*, **50**, 2684 (2017)). The significance of identical structures for comparing the mechanical properties of MOF glasses has been well-recognized by us. Thus, we made intensive attempts to prepare isostructural MOFs with adipate ligands using different metals. However, these efforts were unsuccessful, and they will inevitably remain so because the types of metals (Mg²⁺, Mn²⁺, Co²⁺, and Tb³⁺) have very different in electron configurations and oxidation states, which are decisive factors in coordination geometries and final structures of MOFs. Furthermore, linear aliphatic linkers present challenges in guiding them into a specific structure (*New J. Chem.* **37**, 4130-4139 (2013); *ChemPlusChem* **85**, 845-854 (2020); *Chem. Commun.* **58**, 11402 (2022); *Polymers* **15**, 2891 (2023)). This is contrary to aromatic linkers such as a BDC ligand, which impose structural limitations on the ligand itself, making it relatively straightforward to create an isorecticular framework. Nevertheless, studying the thermal behavior of the four adipate-based MOFs is meaningful to demonstrate the melting and glass transition trends in carboxylate-based MOFs.

According to the reviewer's comments, we have revised the manuscript by incorporating structural details for each C-M-adp MOF and added the analysis of the thermal behavior within the C-M-adp series, guided by trends observed in conventional CP/ZIF glasses. Additionally, we have mentioned the limitations of this analysis in the manuscript, particularly its inability to consider structural factor. The corresponding corrections have been made as shown below.

- Change made:

“Besides the type of metal, various factors can influence the T_m of a MOF. The C-M-adp series features a similar 1D infinite M-O chain as the SBU; however, they do not have an isorecticular structure as carboxylate can have varying binding modes. Specifically, C-M-adp series, including Mg, mostly exhibits a 3D structure, while C-Mn-adp forms a 2D structure where the interconnection of SBUs extends only in the form of a sheet (**Fig. 5a, d, and g**). Since structural influences can be significant in the thermodynamic behavior of MOFs, a direct comparison of thermal trends between these series may be somewhat inaccurate. Nevertheless, the melting behavior of C-Mg-adp and C-Mn-adp can be compared to the tendency that most of the stable liquid CP/MOFs discovered so far contain d¹⁰ metal, suggesting that s-block and d⁵ metals may be novel candidates that could be useful in meltable framework design.¹⁵”

■ Some technical notes.

- For H₂ and CO₂ sorption, the x-axis is the relative pressure (P/P₀) and it is unclear what their P₀ values are.

(Response) We appreciate the reviewer's comments. The corresponding corrections have been made as shown below. We have rewritten 'relative pressure' on the x-axis as 'measured pressure'. The saturation pressure of the gas under each measurement condition is 3.333 MPa for H₂, and 3.4851 MPa for CO₂, which are taken from the NIST website.

● Change made:

C-Mg-adp

G-Mg-adp

Supplementary Fig. 26 Gas sorption isotherm data of C-Mg-adp (top) and G-Mg-adp (bottom) for N₂ (left), H₂ (middle), and CO₂ (right). Filled and hollow circles represent adsorption and desorption, respectively.

- The SEM images in Fig. S5 show that the surface of G-Mg-adp becomes very uneven after heating for 1 h. Any reason?

(Response) We appreciate the reviewer's comment. The uneven surface of G-Mg-adp is caused by vapor evolution from the liquid phase of Mg-adp during heating for 1 hour, as demonstrated in TG-GC-MS results. The evolved CO gas and organic molecules from Mg-adp become trapped as bubbles in the liquid, which has a low fragility index. With prolonged heating, the size of the bubbles increase, forming large cavities in the glass. A similar phenomenon has been reported in long-term heating experiments of ZIF-62 (refer to Figure 5 in *J. Non-Cryst.* **530**, 119806 (2020)). In the revised version, the corresponding figure is Supplementary Fig. 5, and the figure caption has been corrected for clarity as shown below.

- Change made:

Supplementary Fig. 5 SEM images of G-Mg-adp captured after heating C-Mg-adp to its melting point (T_m) at 285 °C with a heating rate of 10 °C min⁻¹. **a**, G-Mg-adp immediately cooled to room temperature once the temperature reached 285 °C. **b**, G-Mg-adp heated at 285 °C for 1 hour and then cooled to room temperature, resulting in the formation of large bubbles due to the evolution of CO gas and organic vesicles.

- What is the stability of the glasses in water/humidity?

(Response) We appreciate the reviewer's comment. To address this concern, we conducted the water stability test of *G*-Mg-adp. The water/humidity stability of *G*-Mg-adp is completely different from it of *C*-Mg-adp. The corresponding results have been added as shown below.

● Change made:

“We observed that *G*-Mg-adp exhibited much higher water stability than *C*-Mg-adp (Supplementary Fig. 29). While *C*-Mg-adp is very soluble in water, *G*-Mg-adp is almost insoluble. As shown in IR spectra, after soaking in water, water-adsorbed *G*-Mg-adp (*G*-Mg-adp-*h*) retains the coordination bond between Mg^{2+} and carboxylate and reverts back after activation at 80 °C in a vacuum.”

Supplementary Fig. 29 Water stability test of *C*-Mg-adp and *G*-Mg-adp. **a**, Photographs for the water treatment processes for *C*-Mg-adp and *G*-Mg-adp powders. Notably, *G*-Mg-adp is not completely dissolved even in a large quantity of water, whereas *C*-Mg-adp dissolves in a small amount of water. **b**, IR data of *C*-Mg-adp, *G*-Mg-adp, *G*-Mg-adp-*h*, and activated *G*-Mg-adp-*h*.

- TGA of pristine C-Mg ADP shows that T_d was evaluated at 330 °C (Fig. S1), while TGA of the obtained glass showed that T_d was around 280 °C (Fig. S2). Can you explain these differences? I think that the marked position of T_d in Fig. S1 is inappropriate because a significant weight change occurs. T_d should be around 280 °C and it is close to the marked T_m (283 °C).

(Response) We appreciate the valuable comments. As the reviewer pointed out, we carefully determined T_d at the onset temperature in the TGA traces for both C-Mg-adp and G-Mg-adp. The newly assigned T_d for C-Mg-adp and G-Mg-adp is 298 °C and 290 °C, respectively. The difference in T_d between C-Mg-adp and G-Mg-adp might be attributed to partial decomposition during melting, which was explained in the response to Comment 2. It is worth noting that the previous TGA results were obtained with a heating rate of 1 °C/min for more precise results. However, since the melting temperature (T_m) was determined based on DSC results, which were measured with a ramp rate of 10 °C/min, we conducted TGA for C-Mg-adp and G-Mg-adp with the same ramp rate. As shown in Review Only Figure R2, the T_d for C-Mg-adp and G-Mg-adp is 320 °C, indicating a significant difference with the T_m (283 °C) of C-Mg-adp. As suggested by Reviewer 2 to conduct measurements for both TGA and DSC with the same ramp rate, we will replace TGA data obtained with 10 °C/min (Supplementary Fig. 1) in the revised manuscript.

Review Only Figure R2. TGA data of **a, b**, C-Mg-adp measured with a ramp rate of 1 and 10 °C/min, respectively, and **c, d**, G-Mg-adp measured with a ramp rate of 1 and 10 °C/min, respectively.

Supplementary Fig 1. TGA data of **a**, C-Mg-adt **b**, G-Mg-adt with 10 °C/min ramp rate. T_d was evaluated at 320 °C, which showed abrupt weight loss on TGA.

Reviewer #2

In this work, the authors investigated the thermal behaviors of a family of rod-spacer type MOFs, which is constructed from a metal carboxylate chain linked by an aliphatic carboxylate ligand of $-(\text{OOC})(\text{CH}_2)_4(\text{COO})-$. With the aim of finding new MOF glasses, the Mg and Mn based MOF seem to be melted and can produce MOF glass upon quenching. The melting and glass forming behaviors are mainly supported by TGA, DSC, PXRD, SEM and other characterization results. In contrast, direct decomposition or amorphization were observed in iso-reticular MOFs with Co and Tb. In deeded, liquid and glass states represent the great advance and interesting new direction of MOFs, considering the different aspects of MOF processing, new MOF function and understanding the nature of glass. Hence, the presented results are valuable for this area and can be considered for publication. However, from the viewpoint of novelty, these findings of melt-quenched carboxylate metal–organic framework glasses are not surprising or beyond expectation. Both melting-quenching and other strategies like mechanical, perturbation methods are proved to be useful to produce glasses, form different types of MOF precursor. In addition, there are several experimental queries on the real glass formation and glass structure. On this occasion, the current manuscript is still too preliminary and major revision is necessary to support its publication.

(Response) We appreciate the reviewer for carefully reading our manuscript and for providing helpful comments. We acknowledge that methods other than melting-quenching are also crucial and useful methods for glass production. Additionally, we agree that several methods are employed in forming MOF glass structures.

Nonetheless, we want to emphasize the significance of discovering and expanding the pool of meltable MOFs. This is essential because these materials exhibit unique behaviors found only within this class. Furthermore, liquid MOFs can significantly enhance our understanding of the dynamic M-L bonds within MOF structures. These research advancements are not only crucial for comprehending the chemical stability of MOFs but also for expanding their potential applications in catalysis, where the dynamics of bonding play a crucial role (*Coord. Chem. Rev.* **496**, 215403 (2023)).

While studies utilizing the unique properties of liquids have been actively explored in the ZIF glass field, research on carboxylate-based MOFs, which constitute the majority of this field, has been limited due to the absence of observation regarding to their liquid phases.

To emphasize the importance of this study, we have strengthened the Introduction section in the revised manuscript as following. We hope that these revisions will convey the impact of our study in this field.

1. Comparing to the inserted 2nd heating DSC which seems normal, the quite flat DSC curve for the first heating run is strange. Have the authors performed some data treatment for the raw signals?

(Response) We appreciate the reviewer's comments. We want to clarify that we did not manipulate the raw data in the DSC measurements. The appearance of a flatter baseline in the DSC curve from the first heating run can be attributed to the strong peak at 180 °C. As demonstrated in Review Only Figure R3, when the DSC curves from the first and second heating runs are displayed on the same Y-axis scale, they exhibit a similar baseline. We believe that the revised Fig. 2b (previously Fig. 1b in the original version), where the DSC curves have been re-plotted on the same scale, will help prevent any misunderstanding among readers.

Review Only Figure R3. DSC curves from the first (left) and second (right) heating runs for C-Mg-adp on the same Y-axis scale.

- Change made:

Figure 2b. DSC curves of C-Mg-adp heated with a ramp rate of 10 °C min⁻¹ under argon, the red line corresponds to the first heating step, and the filled area indicates ΔH_{fus} for C-Mg-adp (134.5 kJ mol⁻¹). The grey line corresponds to the second heating step in the subsequent cycle under the same conditions. The T_m and T_g were evaluated by the onset point of each peak.

2. I noticed a heating rate of 10 K min⁻¹ was used for all the DSC measurements. However, In Supplementary Fig. 1 and S2, the TGA used a much lower heating rate of 1 K min⁻¹. Why the same heating rate are not used. The different heating rate will affect the key parameters of T_m, T_g and T_d, and leads to data mismatching?

(Response) We appreciate the reviewer for their observation and insightful feedback. We are well aware of the impact of heating rates on TGA and DSC results. In typical experiments, it is common practice to use the same heating rate when comparing these results. However, our study intentionally employed different heating rates for TGA and DSC. We did this to achieve more accurate results for the onset decomposition temperature (T_d) in TGA since slower heating rates can reduce temperature delay.

In the case of DSC, our primary objective in this study was to observe the glass transition temperature (T_g). A slow heating rate can significantly affect the signal due to changes in the material's heat capacity, making the accurate observation of T_g challenging. Therefore, it is generally recognized that a faster heating rate can be beneficial for observing T_g. In fact, when we used a heating rate of 1 °C/min for C-Mg-adp, it was very challenging to observe T_g in DSC.

For these reasons, in the original version of the manuscript, we used a heating rate of 1 °C/min for TGA and 10 °C/min for DSC. However, we acknowledge that using different heating rates for TGA and DSC can lead to data discrepancies, which may cause confusion among readers. In response to the reviewer's concern, we have included additional data in this revised paper by measuring TGA at 10 °C/min. The relevant details are provided below.

Review Only Figure R2. TGA data of **a, b**, C-Mg-adp measured with a ramp rate of 1 and 10 °C/min, respectively, and **c, d**, G-Mg-adp measured with a ramp rate of 1 and 10 °C/min, respectively.

- Change made:

Supplementary Fig 1. TGA data of **a**, C-Mg-adp **b**, G-Mg-adp with 10 °C/min ramp rate. T_d was evaluated at 320 °C, which showed abrupt weight loss on TGA.

3. Usually, the glass transition or the DSC curve is repeatable upon heating. However, there is just one DSC curve for the quenched same. It is difficult to estimate the stability of the glass state?

(Response) We appreciate the reviewer's comments. We conducted nine DSC cycling experiments on *G*-Mg-adp and *G*-Mn-adp. The results, as shown below, confirm the presence of a glass transition and the consistent location of the T_g during the heating steps of DSC. This consistency signifies that *G*-Mg-adp and *G*-Mn-adp exhibit reversible and stable glass transition behavior within the temperature range close to their respective T_m .

- Change made: DSC curves of *G*-Mg-adp and *G*-Mn-adp from nine consecutive heating and cooling cycles were added in Supplementary Fig. 30.

Supplementary Fig. 30 DSC curves of **a**, *G*-Mg-adp and **b**, *G*-Mn-adp from nine consecutive heating and cooling cycles. Only the heating step data is displayed to highlight T_g . Measurements were conducted from 50 °C to slightly below each MOF's T_m , with a heating/cooling rate of 10 °C/min.

4. In figure 1c, there is an evident peak at 2theta of 7.5° for G-Mg-adp. How about the measured intensity? If it is a strong diffraction peak, it is somewhat contradictory to a glass state.

(Response) We appreciate the reviewer's comment. The XRPD analysis for G-Mg-adp reveals a broad and weak scattering peak centered around 6-11°. To clearly illustrate the absolute intensity of XRPD patterns for C-Mg-adp and G-Mg-adp, we have presented the unnormalized XRPD data in Review Only Figure R4. The peak intensity of G-Mg-adp is weaker than that of C-Mg-adp by almost 1/100 of a magnitude, which makes it difficult to claim meaningful diffraction in the MOF. Nonetheless, the presence of the broad peak in the glass phase of the MOF can potentially be attributed to the persistence of some structural correlations within the glass, such as Mg-O chain-chain distances. This is because not all coordination bonds break during the melting process in MOF glasses. This phenomenon is commonly observed in ZIF/CP glasses (*Angew. Chem. Int. Ed.* **55**, 5195-5200 (2016); *J. Non-Cryst.* **530**, 119806 (2020); *Nat. Chem.* **13**, 778-785 (2021); *Adv. Func. Mater.* **31**, 2104300 (2021)). Due to the longer ligand length, the broader peak area of G-Mg-adp is visible at a relatively lower angle compared to the ZIF glasses.

Review Only Figure R4. XRPD data of C-Mg-adp (red) and G-Mg-adp (black) with absolute intensity as y-axis. A black arrow indicates the position of the peak for G-Mg-adp.

5. In Supplementary Fig. 3, the glass transition point can not be well determined in the DSC curves with high heating rate above 30 K min⁻¹. Then, the accuracy of fragility is not high.

(Response) We appreciate the reviewer for their observation and acknowledge the error of data. As suggested by the reviewer, we recalculated fragility by using DSC data measured with the heating rate of 10-30 K/min. The resultant fragility is 33, which shows higher R² values in fitting.

● Change made:

- DSC curves for calculating liquid fragility index of *G*-Mg-adp were modified in Supplementary Fig. 2 (previously Supplementary Fig. 3 in the original version).

Supplementary Fig. 2 The determination of liquid fragility index (m) from DSC measurements of *G*-Mg-adp. **a**, DSC data using various heating rates. The calorimetric fictive temperature (T_f) obtained using heating rate of 10 K/min is defined as T_g on $\log(1/q)$ vs. T_g/T_f plot. **b**, Fragilities of *G*-Mg-adp, determined as the of $\log(1/q)$ vs. T_g/T_f plot.

- Modified the manuscript.

“The liquid fragility index (dynamical parameters) of *G*-Mg-adp calculated with T_g obtained from various heating rates (10-30 °C min⁻¹) was 33, which implies that its flowing is hardly observed (**Supplementary Fig. 2**).”

6. The in-suit VT-XRPD data of C-Mg-adp is quite useful and significant evidence to deduce the structure transformation process upon heating. However, the presented data in Supplementary Fig. 4 is too coarse. Especially, if the DSC endothermic peak between 283 and 300 oC is stemming from melting, the PXRD at that temperature range should have no peaks for a liquid state.

(Response) We appreciate the reviewer's comment. We conducted *in-situ* variation temperature XRPD (VT-XRPD) measurements of C-Mg-adp and C-Mn-adp from room temperature at intervals of 20 °C.

- Change made: Added *in-situ* VT-XRPD data of C-Mg-adp and C-Mn-adp.

Supplementary Fig. 5 *in-situ* VT-XRPD data of C-Mg-adp. The data were collected at intervals of 20 °C in the range of 25 °C to 345 °C in the N₂ inert gas atmosphere.

Supplementary Fig. 20 *in-situ* VT-XRPD data of C-Mn-adp. The data were collected at intervals of 20 °C in the range of 25 °C to 285 °C in the N₂ inert gas atmosphere.

7. Another great query is about the local structure analysis of Mg-adp with PDFs. Through the authors give long sentence discussion, for current result, only the short-range atom correlations below 5 angstroms give valuable information. It seems the data are X-ray total scattering data are not well collected, leading to low data resolution at middle or long range. On this case, the analysis for the peak of b, c, d, e, f is meaningless, because we can not definitely say there is or is not a peak.

(Response) We appreciate the reviewer's comment and acknowledge that our PDF data were collected using laboratory-grade equipment, which limits our ability to identify middle-range and long-range order due to the relatively lower Q_{\max} ($\sim 22 \text{ \AA}^{-1}$) and data resolution compared to synchrotron beamline measurements (*J. Am. Chem. Soc.* **145**, 11258-11264 (2023)).

We agree with the reviewer's comment and re-analyzed the PDF data for Mg-adp below 5 \AA . Also, we discovered that the peak previously observed around 0.9 \AA was mainly attributed to background noise. Thus, after re-fitting the data, the assignment of C-H bonds was inevitably removed. We have revised an explanation of the PDF results for the local structure of C-Mg-adp and G-Mg-adp in the revised manuscript.

- Change made:

- Fig. 3 (Fig 2 in the previous version) was modified for clarity, and the explanation of PDFs for C-Mg-adp and G-Mg-adp was revised as below.

“As shown in **Fig. 3b**, peaks 1 and 2 in $G(r)$ correspond to the C–O and C–C bond distances in one adipate ligand, and peak 3 corresponds to the Mg–O coordination bond. Notably, after melt-quenching process, a subtle increase in the bond distance is observed at the position of maximum intensity position for Mg–O (peak 3), while the C–O distance (peak 1) shortens at its peak. These results align with the occurrence of redshift in $\nu(\text{COO}^-)$ IR peaks after melt-quenching, indicating a weakening Mg–O bond strength.³⁴ Peak 4 corresponds well with the unconnected C···C distance and exhibit a slightly increase in G-Mg-adp, suggesting an expansion of the $^- \text{O}_2\text{C}-\text{CH}_2-\text{CH}_2$ bond angle in the adipate ligand during melting. Peak 5 and 6 are associated with C···Mg and Mg···Mg correlation distances, respectively. In a range of $4 - 5 \text{ \AA}$, it is challenging to accurately attribute each peak to its specific origin because there are numerous correlations, but the main contributors are likely to be C···C, C···O, and O···O correlations (**Supplementary Fig. 11**).³⁵ Overall, there is a slight increase in the Mg–O distance during the transformation from C-Mg-adp to G-Mg-adp, but the atomic correlation in local structures remains nearly the same for both. That is, the Mg-adipate connection maintains intact even after melt-quenching in C-Mg-adp.”

Fig. 3 | Local structure analysis of Mg-adp with PDFs. a, $G(r)$ for samples of C-Mg-adp (red) and G-Mg-adp (black). **b**, Local coordination environments of Mg-adp. To represent various bonding modes of adipates with similar correlation distances, the half fragment of one adipate with $\mu^3-\eta^1:\eta^2$ mode is shown. 1-6 in the figures represent the bonds and correlations between two atoms, respectively. Color scheme: C, gray; O, red; H, white; and Mg, light green.

8. Comparing to the Mg-MOF, the characterization for Mn-adp is far from sufficient. We can not deduce what kind of structure transformation occurs, based on presented limited result. In Supplementary Fig. 17, the $T_m = 238$ Oc is given for C-Mn-adp, how about the standard to estimate melting in the continuous DSC curves?

(Response) We appreciate the reviewer's comments and acknowledge the limitations of the characterization of Mn-adp in our initial manuscript. Thus, we conducted comprehensive analyses on Mn-adp, including continuous and cycling DSC, *in-situ* VT-XPRD, XAFS, TG-GC-MS, FT-IR spectra, and NMR.

Regarding the determination of melting points in continuous DSC, we adhere to a widely accepted procedure of identifying the onset point of sharp endothermic peaks accompanied by significant enthalpy changes as T_m (*Methods Cell Biol.* **84**, 115-114 (2008); *Methods for Phase Diagram Determination*, Elsevier (2011); *Chem. Rev.* **119**, 7848-7939 (2019)).

In the case of Mn-adp, due to the desolvation of the coordinating solvent occurring just before melting, the clear observation of the change in gradient indicates the presence of an endothermic reaction at that temperature, distinct from desolvation. Thus, we defined the onset point as T_m based on the corresponding change in peak gradient during melting.

We also investigated the occurrence of glass transition in consecutive DSC runs at varying maximum temperatures of 200 °C, 220 °C, 240 °C, and 430 °C with subsequent cooling steps.

- Change made:
 - Continuous and cycling DSC, *in-situ* VT-XPRD, XAFS, TG-GC-MS, FT- IR spectra, and NMR for Mn-adp were added in Supplementary Fig. 19, 20, 21, 22, 23 and 30, respectively.
 - Added the consecutive DSC for C-Mn-adp.

Supplementary Fig. 19 a, The DSC curves of C-Mn-adp depicts the heating, cooling, and subsequent reheating cycles. Both heating and cooling processes were carried out at a ramping rate of 10 °C/min, with the initial heating phase reaching 250 °C. **b**, Consecutive DSC runs with progressively higher maximum temperatures: 200°C, 220°C, 240°C and 430°C. Upon reaching each specified maximum temperature, C-Mn-adp was maintained under isothermal conditions at that temperature for 5 minutes, followed by a cooling step. A glass transition of Mn-adp can be observed only after reaching at least 240°C, as indicated by the 4th curve at 430°C.

Supplementary Fig. 20 VT-XRPD data of C-Mn-adp. The data was measured at 20 °C intervals within the temperature range of 25 – 285 °C in the N₂ inert gas atmosphere.

Supplementary Fig. 21 The wavelet contour using Morlet method with $\eta = 10$ and $\sigma = 1$ over XAFS data of **a**, C-Mn-adp and **b**, G-Mn-adp. **c**, The Fourier-transforms of XAFS spectra as a function of radial distance for C-Mn-adp (red) and G-Mn-adp (black). Phase shift within the data set was uncorrected.

Supplementary Fig. 22 a, TG-GC-MS data were collected for C-Mn-adp by heating up to 240 °C, followed by isothermal measurements for 3 h. The types of species detected for each range were represented at the top. **b**, The FT-IR data collected after the heat treatment, compared with C-Mn-adp and G-Mn-adp.

Supplementary Fig. 23 ¹H NMR spectra for C-Mn-adp, G-Mn-adp and the sample after heat treatment at 285 °C for 3 h.

Supplementary Fig. 30 DSC curves of **a**, *G*-Mg-adp and **b**, *G*-Mn-adp from nine consecutive heating and cooling cycles. Only the heating step data is displayed to highlight T_g . Measurements were conducted from 50 °C to slightly below each MOF's T_m , with a heating/cooling rate of 10 °C/min.

9. The relevance between liquid/glass forming and chemical/structure character is essential and important for MOF glass research. In this work, the Mg and Mn MOF behave in different fashion, comparing to the Co and Tb compound. However, the presented explanation is too general without data supports.

(Response) We appreciate the reviewer's comment and acknowledge that this could be a point of concern. According to the reviewer's comments, we have revised the manuscript by incorporating structural details for each *C*-M-adp MOF and added the analysis of the thermal behavior within the *C*-M-adp series, guided by trends observed in conventional CP/ZIF glasses.

- Change made:

- We have added the X-ray structures of *C*-Mn-adp, *C*-Co-adp, and *C*-Tb-adp in Fig. 5.

Fig. 5 | Variations in the thermal transition of M-adp ($M = \text{Mn}^{2+}$, Co^{2+} , Tb^{3+}). Single-crystal structures of M-adp (**a**, **d**, and **g**), OM images of C-M-adp (**b**, **e**, and **h**), and OM images of C-M-adp subsequent to thermal conversion in an inert gas (**c**, **f**, and **i**). C-Mn-adp could form G-Mn-adp through a melt-quenching process (**c**), while C-Co-adp undergoes amorphization by desolvation (**f**), and C-Tb-adp were maintained the morphology of the crystal until calcination (**i**).

- We have added explanation on the the X-ray structures of C-Mn-adp, C-Co-adp, and C-Tb-adp in Fig. 5.

“Besides the type of metal, various factors can influence the T_m of a MOF. The C-M-adp series features a similar 1D infinite M-O chain as the SBU; however, they do not have an isorecticular structure as carboxylate can have varying binding modes. Specifically, C-M-adp series, including Mg, mostly exhibits a 3D structure, while C-Mn-adp forms a 2D structure where the interconnection of SBUs extends only in the form of a sheet (**Fig. 5a, d, and g**). Since structural influences can be significant in the thermodynamic behavior of MOFs, a direct comparison of thermal trends between these series may be somewhat inaccurate. Nevertheless, the melting behavior of C-Mg-adp and C-Mn-adp can be compared to the tendency that most of the stable liquid CP/MOFs discovered so far contain d^{10} metal, suggesting that s-block and d^5 metals may be novel candidates that could be useful in melttable framework design.¹⁵”

10. The authors have pointed out the low fragility index of 26.3 implies that the flowing is hardly observed for the MOF glass. But there is a stalactite-like and puffed shapes were found in Supplementary Fig. 16 for G-Mg-adp. How such a image was obtained?

(Response) We appreciate the reviewer's comment. The two images were obtained through an annealing process at 265 °C (slightly above its T_g) for a duration of 12 hours after reaching the T_m of C-Mg-adp.

We acknowledge the previously inadequate elucidation of the experimental methodologies presented in our manuscript and supplementary information.

- Change made:

For the manuscript, “Furthermore, the presence of spike-like and puffed shapes observed during annealing above T_g confirms the transition of Mg-adp into a viscous liquid state (**Supplementary Fig. 5 and 6**).”

For the supplementary data, “**Supplementary Fig. 6** SEM images of G-Mg-adp, obtained after quenching and subsequent annealing at 265 °C for 12 hours. Stalactite-like and puffed shapes are occurred from viscous fluid, suggesting a melted phase of Mg-adp”.

11. On the whole, the data quality, organization and presentation are poor, additional experiment and careful manuscript revision are absolutely necessary. For example, in Supplementary Fig. 5, “SEM images of G-Mg-adp was heated at T_m for 0 minutes (top) and 1 hour (bottom)”, what is the meaning for these data? Similar, in Supplementary Fig. 19, 0 minutes? In Supplementary Fig. 15, what is the exact treatment temperature of each samples?

(Response) We appreciate the constructive feedback from the reviewer. During the revision process, we have tried to enhance the quality of the figures and to provide detailed explanations for the data throughout the manuscript. For example, we have modified the text below in response to review’s concerns.

- Change made:

For the **Supplementary Fig. 5**, “SEM images of *G*-Mg-adp captured after heating *C*-Mg-adp to its melting point (T_m) at 285 °C with a heating rate of 10 °C min⁻¹. **a**, *G*-Mg-adp immediately cooled to room temperature once the temperature reached 285 °C. **b**, *G*-Mg-adp heated at 285 °C for 1 hour and then cooled to room temperature, resulting in the formation of large bubbles due to the evolution of CO gas and organic vesicles. “

For the **Supplementary Fig. 16** (previous Supplementary Fig. 15), “XRPD data of *C*-M-adp (M: Mn²⁺, Co²⁺, Tb³⁺), *G*-Mn-adp and amorphized Co-adp. *G*-Mn-adp was yielded through the melt-quenching process of *C*-Mn-adp under inert gas atmosphere at 240 °C for 10 minutes. The amorphization of *C*-Co-adp was conducted in the inert gas at 200 °C for 10 minutes.”

For the **Supplementary Fig. 24** (previous Supplementary Fig. 19), “SEM images of *G*-Mn-adp subjected to quenching immediately after reaching its T_m .”

Reviewer #3

(1) The presented studies provide a very first example of a carboxylate-based MOF glass. In general, the reviewer recommends the publication of this paper after addressing certain issues that have been discussed below. The reviewer acknowledges and agrees with the authors' statement that this is the first example of its kind. However, it remains unclear why this particular system had not been prepared earlier, considering that carboxylate-containing MOFs are one of the most extensively investigated classes. While the reviewer is not debating that it is a first example, it is important to emphasize in the introduction why these studies could be useful.

(Response) We appreciate the reviewer for their time and positive feedback. We want to emphasize the significance of discovering and expanding the pool of meltable MOFs. This is essential because these materials exhibit unique behaviors found only within this class. Furthermore, liquid MOFs can significantly enhance our understanding of the dynamic M-L bonds within MOF structures. These research advancements are not only crucial for comprehending the chemical stability of MOFs but also for expanding their potential applications in catalysis, where the dynamics of bonding play a crucial role (*Coord. Chem. Rev.* **496**, 215403 (2023)). While studies utilizing the unique properties of liquids have been actively explored in the ZIF glass field, research on carboxylate-based MOFs, which constitute the majority of this field, has been limited due to the absence of observation regarding to their liquid phases. To emphasize the importance of this study, we have strengthened the Introduction section in the revised manuscript as following. We hope that these revisions will convey the impact of our study in this field.

- Change made: We have strengthened the Introduction section as below.

“Introduction

Metal–organic frameworks (MOFs) are coordination networks with potential pores in a well-ordered structure composed of metal ions and polydentate organic ligands.¹ Over the past few decades, the field of MOFs has significantly expanded because of their high designability and tunability.² Despite their various properties, the practical applications of MOFs are limited because of their crystalline powder nature and low processability.^{3,4} To overcome the limitations of crystalline MOFs, efforts have been made to transform MOFs into more practical and versatile forms and shapes through integration with polymers, palletization, and processing into beads.⁵ Among these approaches, meltable MOFs have recently gained significant attention, as their liquid phase allows for molding without dependence on other materials.⁶ Moreover, molten MOFs can generate a novel type of material, MOF glasses via a melt-quenching process.^{7,8} These glass structures retain the components of the original crystal and exhibit unique properties such as a monolithic manner⁹, enhanced ion conductivity⁹, transparency¹⁰, and luminescence¹¹. They also have a distorted pore network distinct from the mother MOFs.¹²

To enable melting in a MOF, the MOF must have either a low melting temperature (T_m) or a high thermal decomposition temperature (T_d) to satisfy the condition, $T_m < T_d$ (**Fig. 1**). This requirement arises from the fundamental concern that the average local coordination environment of the structures must be maintained while their long-range order is lost.^{13,14} So far, studies on meltable MOFs have mostly focused on zeolitic-imidazole frameworks (ZIFs)

with high T_d owing to their thermally stable azole ligands, and coordination polymers (CPs) composed of phosphates, amides, and sulfonates, which form weak coordination bonds with metals, thereby lowering the T_m of the framework.^{15,16}

Fig. 1 | Schematic graphics illustrating the principle of MOF melting. For the MOF to melt, partial decooordination of the metal-ligand bonds must occur before the thermal decomposition of framework ($T_m < T_d$). Therefore, a meltable MOF can be designed by controlling T_m and T_d .

Despite recent advancements in MOF glass field, an important area that still need attention is the melting and vitrification of carboxylate-based MOFs, which constitute a significant majority of MOFs.¹⁷ Most carboxylate-based MOFs decompose before vitrification owing to the strong bonds between carboxylate and metal centers, which elevate the T_m of the framework above its T_d .^{15,18} Extensive research has been conducted on carboxylate-based MOFs to date; however, to the best of our knowledge, no melting behavior has been reported among MOFs whose structures use purely carboxylates as ligands. Considering the active research on thermally decomposing MOFs to create nanomaterials, such as metal oxides and carbon, the absence of meltable carboxylate-based MOFs becomes even more surprising¹⁹. Since the discovery of the liquid and glassy states of MOFs is relatively recent, it is possible that earlier researchers did not observe or recognize these states in carboxylate-based MOFs. Another possible factor is the prevalent strategies in reticular design. These strategies commonly involve using aromatic ligands to create well-ordered structures and metals with strong coordination bonds to form robust frameworks, resulting in higher crystallinity, porosity, and good stability in MOFs.^{20,21} However, aromatic ligands have few conformations due to their rigid local structure and high symmetry properties, which do not provide entropic benefits for the melting of MOFs.^{13,22} Furthermore, the strong metal-ligand bond directly contributes to increasing the MOF's T_m , thus hindering its melting.

We introduce a novel class of meltable carboxylate-based MOFs consisting of Mg^{2+} or Mn^{2+} ions and an *aliphatic* carboxylate linker, adipate (adp, $\text{-(OOC)(CH}_2\text{)}_4\text{(COO)-}$). Compared to aromatic carboxylate ligands, aliphatic carboxylate ligands have lower thermal stability and higher degree of conformational freedom. The d^0 and d^5 configuration of metals may lead to a reduction in the thermal energy required for the breaking of metal-ligand bonds, owing to their low crystal field stabilization energy (CFSE), which is similar to d^{10} metals found in existing CP/ZIF glasses.^{7-8,10,15} Additionally, the aliphatic carboxylate amplifies the gap in rotational entropy between the solid and liquid phases of MOFs when compared to an aromatic linker, due to its ability to adopt numerous conformations in the liquid phase.^{15,22} Simultaneously, the aliphatic linker may reduce the thermal stability of the framework. Based on these properties, we have previously demonstrated the thermal conversion of aliphatic ligand-based MOFs with low T_d into hierarchically nanoporous metal oxides with nanocrystalline frameworks.²³

Recently, there have been a few reports on the synthesis of carboxylate-based MOF glasses.²⁴⁻²⁶ However, these studies differ from the present work in that the starting materials for glasses are hydrogen-bonded networks of the metal complexes or amorphous coordination compounds. Certainly, adopting this bottom-up approach is a compelling strategy for synthesizing MOF glasses. These methods not only offer a facile synthesis process with substantial industrial potential but also the capability to vitrify thermally unstable frameworks. Nevertheless, achieving a molten phase in MOFs remains a notable challenge due to its potential to unlock unprecedented applications distinct from non-meltable frameworks. The presence of a liquid phase in MOFs introduces captivating possibilities, including the potential for novel structural evolution²⁷, the creation of eutectic composites with other substances²⁸, and the utilization of MOFs as matrix materials themselves²⁹, contrasting with the conventional application of MOFs as solid additives³⁰. Moreover, a fundamentally understanding of meltable carboxylate-based MOFs may assist future studies aimed at imparting melt behavior to existing MOFs that have non-meltable properties.

While the bottom-up approach circumvents the thermodynamic challenges of carboxylate-based frameworks, the absence of a solid-liquid phase transition or T_m in crystalline carboxylate frameworks²⁶ restricts the variety of reported liquid/glass MOFs. Consequently, it has impeded the establishment of rational design principles for meltable MOF structures.

Here, we utilize the low T_m of the crystalline MOFs ($[\text{Mg}_4(\text{adipate})_4(\text{DMA})(\text{H}_2\text{O})] = \text{C-Mg-adp}$ and $[\text{Mn}_2(\text{adipate})_2(\text{DMA})] = \text{C-Mn-adp}$) by controlling the enthalpy of fusion (ΔH_{fus}) and the entropy of fusion (ΔS_{fus}) to trigger their thermal transition into the liquid phase and eventually create the carboxylate-based MOF glasses ($G\text{-Mg-adp}$ and $G\text{-Mn-adp}$, respectively). These adipate MOFs exhibit a high glass-forming ability (GFA), signifying their capacity to easily vitrify in their liquid state while preventing recrystallization. X-ray total scattering data and pair distribution functions (PDFs) confirmed that $G\text{-Mg-adp}$ retains the connectivity between the carboxylate and metal ions. The mechanical properties of $G\text{-Mg-adp}$ were characterized using nanoindentation and exhibited higher hardness (H) and elastic modulus (E) than those of the reported CP glasses.”

(2) The reviewer also suggests including the table of known the imidazole-containing MOF glasses with the relevant porosity and surface area. The reviewer commends the authors on their thorough analysis, but recommends comparing the findings, whenever possible, with existing ZIF-based glass systems. While some tables are already provided and prove to be highly useful, they alone may not be sufficient in certain cases.

(Response) We appreciate the reviewer’s advice and carefully considered the suggestions, particularly in comparing the porosity of our samples with ZIF-based glass. Unfortunately, our analysis indicated that $G\text{-Mg-adp}$ exhibits almost non-porous properties. Consequently, the summarized table for porosity did not provide meaningful insights.

However, while organizing the table, we encountered a recent article that delves into an in-depth study on the porosity of ZIF glass. (Refer to ‘Quantification of gas-accessible microporosity in metal-organic framework glasses’, *Nat. Commun.* **13**, 7750 (2022).) This article sheds light on the porosity determination in MOF glass from various perspectives, offering a more comprehensive analysis than our table.

(3) The Reviewer recommends improving the quality of the illustrative content in the paper. The figures appear to have a relatively low quality, with instances of font mismatching between the figures and the main text, which hinders the clarity of certain labels.

(Response) We appreciate the reviewer's comment. We have made efforts to harmonize the textual content with the illustration, aiming to enhance the overall quality of the paper. We also apologize for the subpar presentation quality in the earlier version of our manuscript.

- Change made:
 - The figures in both the manuscript and supplementary materials have been redesigned for enhanced clarity throughout the manuscript.
 - The fonts in the text and figures have been adjusted for consistency.

(4) Could the authors re-check the fit (PDF analysis) presented in Figure 2?

(Response) We appreciate the reviewer's comment. Following the reviewer's advice, we re-checked the PDF analysis in Fig. 3 (Fig. 2a in the previous version).

The modified PDF analysis considers only short-range orders ($r < 5 \text{ \AA}$) as reliable values, considering the Q_{\max} value and resolution of lab-scale equipment. Also, the peak appeared around 0.9 \AA was the result of background noise, and so the previous assignment of this peak as C-H bonds has been deleted. In addition, we added explanation of the PDF results for the local structure of *C*-Mg-adp and *G*-Mg-adp. $G(r)$ revealed that the local coordination environments ($r < 5 \text{ \AA}$) of *G*-Mg-adp are nearly identical to those of *C*-Mg-adp. The differences in peaks attributed to Mg-O, C-O and C-C were explained in the main text. Especially, unconnected C-C distances can be varied upon melting due to the freely rotatable C-C single bonds in the aliphatic linker. This phenomenon is also observed in coordination polymers composed of aliphatic amide. (refer to Supplementary Fig. 39 in *J. Am. Chem. Soc.* **143**, 7, 2801–2811 (2021).)

- Change made:
 - Fig. 3 (Fig 2 in the previous version) was modified for clarity, and the explanation of PDFs for *C*-Mg-adp and *G*-Mg-adp was revised as below.

“As shown in **Fig. 3b**, peaks 1 and 2 in $G(r)$ correspond to the C-O and C-C bond distances in one adipate ligand, and peak 3 corresponds to the Mg-O coordination bond. Notably, after melt-quenching process, a subtle increase in the bond distance is observed at the position of maximum intensity position for Mg-O (peak 3), while the C-O distance (peak 1) shortens at its peak. These results align with the occurrence of redshift in $\nu(\text{COO}^-)$ IR peaks after melt-quenching, indicating a weakening Mg-O bond strength.³⁴ Peak 4 corresponds well with the unconnected C-C distance and exhibit a slightly increase in *G*-Mg-adp, suggesting an expansion of the $\text{O}_2\text{C-CH}_2\text{-CH}_2$ bond angle in the adipate ligand during melting. Peak 5 and 6 are associated with C-Mg and Mg-Mg correlation distances, respectively. In a range of 4 – 5 \AA , it is challenging to accurately attribute each peak to its specific origin because there are

numerous correlations, but the main contributors are likely to be C...C, C...O, and O...O correlations (**Supplementary Fig. 11**).³⁵ Overall, there is a slight increase in the Mg-O distance during the transformation from C-Mg-adp to G-Mg-adp, but the atomic correlation in local structures remains nearly the same for both. That is, the Mg-adipate connection maintains intact even after melt-quenching in C-Mg-adp.”

Fig. 3 | Local structure analysis of Mg-adp with PDFs. a, $G(r)$ for samples of C-Mg-adp (red) and G-Mg-adp (black). **b,** Local coordination environments of Mg-adp. To represent various bonding modes of adipates with similar correlation distances, the half fragment of one adipate with $\mu^3-\eta^1:\eta^2$ mode is shown. 1-6 in the figures represent the bonds and correlations between two atoms, respectively. Color scheme: C, gray; O, red; H, white; and Mg, light green.

(5) It is not clear to the reviewer what is intended to be observed based solely on the presented figures? What parts of the presented pictures are important? The reviewer suggests modifying the main text and the corresponding figure caption.

We appreciate the reviewer for their advice. The pictures were modified to improve clarity and ensure it accurately conveys the intended contents.

- Change made:

- Lines or arrows were incorporated within the figure to enhance clarity.
- The manuscript and figure captions have been revised to better correspond with the figures.

Reviewers' Comments:

Reviewer #1:

Remarks to the Author:

The authors have carefully addressed all the points and questions raised by the reviewers. Many additional experiments and revisions of the discussion have been carried out, resulting in an even better manuscript. The melting of MOFs composed of dicarboxylate is still new and brings high novelty and impact.

However, my main scientific concern is that the modulus (and hardness) is higher than in the ZIF system: the theoretical calculations carried out in the revision showed that the binding energy is carboxylate-metal ion > imidazole-metal ion, but this is obvious and only gives information on the local molecular scale geometry. The question to be addressed here is to determine from several experiments why the resulting bulk glass has such an anomalously high modulus. The authors state in point (1) of the revision letter that the crystals are very soft and their experiments, including PXRD, showed that the crystal structures are easily transformed to another phase. This structural flexibility, mainly due to the aliphatic part of the adipate, may contribute to a significant reduction of the Young's modulus in bulk glass. Could the validity or anomaly be better confirmed, for example, by measuring fracture toughness, which is always correlated with Young's modulus (e.g. Nat Commun 2020 Vol. 11 Issue 1 Pages 2593)? In any case, this mechanical property is still hard to believe based on the composition and structure of the MOF, and further experimental evaluation seems essential.

The revised Figure S5 concludes that the gas released during heating is CO, but where is the evidence for this? Without experimental evidence, it is not recommended to state this explicitly.

The resistance to water has been done in Figure S29. The question we want to ask is not only about solubility, but also about changes in glass structure and properties: we know that G-Mg-adp does not dissolve when immersed in water, but does it also retain most of its properties as a pristine glass? Or does it undergo some disturbance such as partial crystallisation?

As the author states, solid state NMR of Mg is difficult, but XAS of Mg can be done with synchrotron soft X-rays. It is unfortunate that this has not been done.

Reviewer #2:

Remarks to the Author:

Based on additional experiment results and analysis, the reviewer agrees the quality of the whole paper further improved. Many queries from three independent reviewers have been answered in different degree. The authors' great effort is praisable, but there are still some mistakes or unclear points that should be carefully revised, before a final acceptance for publication.

1. For glass study, the authors should pay particular attention to thermal analysis like TGA and DSC. However, mistakes still exist for this revised version.

1) In Fig. S1a, the starting point for the TG curves should be 100%. Similar problem was found in Figure S17 for c-Mn-adp.

2) In Fig. S1b, the TG curves for G-Mg-adp is in correct. There is a strange and large weight loss about 20% below 200 oC, but we all know there is no existed solvent for the quenched sample from high-temperature melt.

3) For the same sample and repeated measurements, why there is such large shift in DSC curves for G-Mg-adp and G-Mn-adp and? See Fig. S30.

4) All the DSC should list the direction of endothermal or exothermal.

2. The statement of water stability in Figure S29 is in-correct. A MOF can not be dissolved in solution, and the exact phenomena for C-Mg-adp is decomposition.

3. Considering the new EXAFS, PDF and water solubility data, it points to a clear possibility that the glassy state has a larger Mg cluster compared to the crystal state. Could the authors provide ICP and elemental analysis data to see whether the linker ratio has changed?

4. Related to the previous question, can the glassy MOF be restored to a crystalline phase by any method (heating, solvent annealing, etc)? It also shows whether the metal cluster has fundamentally changed.

5. Though the resolution of lab-scale equipment is limited, the raw results of middle and long range correlation should be presented (maybe in SI) rather than deleted.

6. As reviewer 1 pointed out, the papers 10.1021/jacs.2c04532, 10.26434/chemrxiv-2023-05kv3, have reported that MOFs composed of ligands with one carboxylate group transform to a glassy state through a perturbation method. But the authors didn't mention or cite this work. I think a proper way is adding essential discussion and comparison in the introduction part, so as to clearly show the history of carboxylate-based MOF glasses and highlight the new advance of the presented work.

Reviewer #3:

Remarks to the Author:

The authors addressed the reviewer's concerns. Therefore, the reviewer recommends the paper for acceptance.

Reviewer #1 (Remarks to the Author):

The authors have carefully addressed all the points and questions raised by the reviewers. Many additional experiments and revisions of the discussion have been carried out, resulting in an even better manuscript. The melting of MOFs composed of dicarboxylate is still new and brings high novelty and impact.

We are grateful to the reviewer for their various suggestions that provided insight into our research from various perspectives.

1. However, my main scientific concern is that the modulus (and hardness) is higher than in the ZIF system: the theoretical calculations carried out in the revision showed that the binding energy is carboxylate-metal ion > imidazole-metal ion, but this is obvious and only gives information on the local molecular scale geometry. The question to be addressed here is to determine from several experiments why the resulting bulk glass has such an anomalously high modulus. The authors state in point (1) of the revision letter that the crystals are very soft and their experiments, including PXRD, showed that the crystal structures are easily transformed to another phase. This structural flexibility, mainly due to the aliphatic part of the adipate, may contribute to a significant reduction of the Young's modulus in bulk glass. Could the validity or anomaly be better confirmed, for example, by measuring fracture toughness, which is always correlated with Young's modulus (e.g. Nat Commun 2020 Vol. 11 Issue 1 Pages 2593)? In any case, this mechanical property is still hard to believe based on the composition and structure of the MOF, and further experimental evaluation seems essential.

(Response) I am truly grateful to the reviewer for providing comments that encourage us to contemplate our findings more deeply.

I understand that accepting the unprecedented mechanical properties of *G*-Mg-adp can be challenging because its tendency differs from that of ZIF glasses, which have been extensively studied in this field. We should have provided a more convincing explanation, and I apologize for not doing so in the previous revision.

First, it should be noted that as you suggested, fracture toughness can have positive correlation with Young's modulus (E) and hardness (H), especially terms of H/E and H^3/E^2 , but this is not generally applicable as discussed in the paper reported by Chung et al ('Commentary on using H/E and H^3/E^2 as proxies for fracture toughness of hard coatings', *Thin Solid Films*, **2019**, 688, 137265). Also, I know that the reviewer's point is the origin of the high mechanical properties in *G*-Mg-adp, not its fracture toughness itself.

In our earlier explanation of the mechanical properties of *G*-Mg-adp, we primarily focused on the strong coordination bond within it. Additionally, we now recognize that the deformed status of *G*-Mg-adp upon melting is a crucial factor influencing its mechanical properties. *C*-Mg-adp exhibits flexible structural transformations attributed to the aliphatic part of the adipate. The question arises as to whether this flexibility of adipate ligands in *C*-Mg-adp can be reflected in the softness of *G*-

Mg-adp. However, our claim is the opposite: high degree of deformation in Mg-adp during melting process results in the entanglement of the structure via the intermolecular interactions between aliphatic parts as well as the strong M-L bonds. Our argument is based on the following.

- (1) As evident from the preceding Supplementary Fig. 28, Mg-adp exhibits a higher ΔH_{fus} compared to previously reported MOFs. This indicates that during the melting process, the network structure undergoes more significant deformation than existing MOFs, resulting in a liquid phase that allows for greater freedom of conformation on the network of glass. In contrast, the network of ZIF exhibits minimal deformation in its network structure with an extremely low ΔH_{fus} . Therefore, in Mg-adp, a network with entanglement can be introduced into its glass state during the melt-quenching process, which enhance its mechanical properties.
- (2) The non-porous nature of *G*-Mg-adp can be attributed to the significant network deformation described above. As porosity increases, packing density decreases, and it is generally known that porosity and modulus are inversely proportional. This phenomenon has been cited as one of the reasons for the low mechanical properties observed in existing ZIF glasses in previous research (*Proc. Natl. Acad. Sci. U. S. A.* **117**, 10149-10154 (2020)).
- (3) If we look at *C*-Mg-adp in terms of a coordination polymer composed of aliphatic chains, it is worthy of comparison with organic polymers. For example, polyethylene is a semicrystalline polymer containing long $-\text{CH}_2-$ chains, and its packing density and orientation highly affect many of the physical properties such as tensile strength, stiffness, and hardness. It means that the structure and intermolecular structure in the resulting product are much more crucial rather than the ease of conformational change in building blocks directly responsible for the production of soft materials.
- (4) As suggested by the theoretical calculation in Supplementary Table 4, the bulk modulus of *C*-Mg-adp (13.2 GPa) is smaller than that of ZIF-4 (29.9 GPa). After vitrification, the trend is reversed. Therefore, while the introduction of aliphatic ligands in crystalline MOF construction affects the structural flexibility and melting/decomposition temperature of the MOF itself, the flexible ligand during vitrification results in an even stronger melt-quenched MOF glass. This is because during the melting process the entire structures are dramatically changed to the nonporous and dense phase.

We have provided a more in-depth explanation of the factors contributing to the anomalous mechanical properties of *G*-Mg-adp in the discussion section.

- Change made:
 - Modified the mechanical properties of *G*-Mg-adp part of manuscript.

This result can be interpreted, with caution, as being caused by both the highly deformed network structure in *G*-Mg-adp (see the discussion section below) and the stronger coordination bond of *G*-Mg-adp compared to ZIF glasses³⁸.

- Modified the discussion part of manuscript.

This feature is due, in part, to the relatively strong coordinate bonds, which partially contribute to the stabilization of local structure in the molten phase of MOFs, resulting in a less fragile liquid⁵⁹⁻⁶². In addition, the substantial flexibility of the aliphatic ligands in Mg-adp enables a high degree of deformation in C-Mg-adp structure, leading to the formation of a nonporous and dense phase upon the melting process. Since modulus and hardness are inversely proportional to porosity^{63,64}, this structural deformation is one of the factors contributing to the harder mechanical properties of G-Mg-adp compared to ZIF glasses, which have microporosity after vitrification⁶⁵.

2. The revised Figure S5 concludes that the gas released during heating is CO, but where is the evidence for this? Without experimental evidence, it is not recommended to state this explicitly.

(Response) We appreciate the reviewer's comment. We identified the presence of CO through the peak at $m/z=28$ in TG-GC-MS data, as mentioned in Supplementary Fig. 9a. During a specific period in the measurement, the $m/z=28$ peak consistently emerged as the predominant peak, indicating its nature as a molecular species rather than an ionic one. Possible molecular species for $m/z=28$ include N_2 , CO, and C_2H_4 . The reason for identifying the peak as CO is twofold: Firstly, the use of an Ar atmosphere allows us to exclude the influence of N_2 flow gas. Secondly, upon excluding $m/z=28$ from C_2H_4 , the expected $m/z=27$ peak with the highest intensity did not appear throughout the entire measurement. Additional information on atmospheric composition is included in the captions for previous Supplementary Fig. 9 and 23.

- Change made: Provided detailed conditions for TG-GC-MS measurements in the captions of previous Supplementary Fig. 9 and 23.

Supplementary Fig. 9 a, TG-GC-MS data collected for C-Mg-adp by heating up to 285 °C, followed by isothermal measurements for 3 h **under an Ar atmosphere**. The types of species detected for each range were represented at the top. **b**, The FT-IR data collected after the heat treatment of Mg-adp, compared with C-Mg-adp and G-Mg-adp.

Supplementary Fig. 23 a, TG-GC-MS data were collected for C-Mn-adp by heating up to 240 °C, followed by isothermal measurements for 3 h **under an Ar atmosphere**. The types of species detected for each range were represented at the top. **b**, The FT-IR data collected after the heat treatment, compared with C-Mn-adp and G-Mn-adp.

3. The resistance to water has been done in Figure S29. The question we want to ask is not only about solubility, but also about changes in glass structure and properties: we know that G-Mg-adp does not dissolve when immersed in water, but does it also retain most of its properties as a pristine glass? Or does it undergo some disturbance such as partial crystallisation?

(Response) We appreciate the reviewer's comments. To verify the properties of *G*-Mg-adp after immersion in water, we provided XRPD data for Supplementary Fig. 30. The XRPD data for *G*-Mg-adp soaked in water were measured after collecting the sample by filtration, designated as *G*-Mg-adp-*h*. Subsequently, they were activated in a vacuum oven at 80 °C for 5 h and the collected solid (activated *G*-Mg-adp-*h*) was measured again. Although the structural relationship among the pristine glass, *G*-Mg-adp-*h* and its activated form is unclear, XRPD data indicate that all samples remain an amorphous state.

- Change made: Incorporated the XRPD data in Supplementary Fig. 30 (Supplementary Fig. 29 in the previous version) and an explanation of this data has been included in the manuscript.

Supplementary Fig. 30 Water stability test of *C*-Mg-adp and *G*-Mg-adp. **a**, Photographs for the water treatment processes for *C*-Mg-adp and *G*-Mg-adp powders. Notably, *G*-Mg-adp remains stable even in a large quantity of water, whereas *C*-Mg-adp undergoes degradation in a small amount of water. **b**, FT-IR data of *C*-Mg-adp, *G*-Mg-adp, *G*-Mg-adp-*h*, and activated *G*-Mg-adp-*h*. **c**, XRPD data of *G*-Mg-adp, *G*-Mg-adp-*h*, and activated *G*-Mg-adp-*h*.

“While *C*-Mg-adp rapidly degrades in a small amount of water, *G*-Mg-adp almost retains its shape and remains immersed even with an excess quantity of water. As shown in the IR spectra, after soaking in water, the filtered *G*-Mg-adp (*G*-Mg-adp-*h*) retains the coordination bond between Mg²⁺ and carboxylate and reverts back after activation at 80 °C in a vacuum. Throughout the entire process, *G*-Mg-adp consistently maintains an amorphous states and does not undergo crystallization. The origin of the improved water stability in MOF through vitrification may be revealed through detailed structural analysis of *G*-Mg-adp, *G*-Mg-adp-*h*, and activated *G*-Mg-adp-*h*. However, a complete survey of these topics is beyond the scope of this paper.”

4. As the author states, solid state NMR of Mg is difficult, but XAS of Mg can be done with synchrotron soft X-rays. It is unfortunate that this has not been done.

(Response) We appreciate and fully agree with the reviewer’s comments. The studies mentioned in previous letters have reported Mg EXAFS utilizing synchrotron soft X-ray. (*Am. Mineral.* **93**, 495-498 (2008). *Chem. Mater.* **22**, 161-166 (2010)). Unfortunately, challenges in reserving the necessary beamline hindered us from conducting this analysis, and even the beam line in Korea has been closed for a while, making the situation more difficult. We sincerely regret this limitation and look forward to exploring this aspect more thoroughly in our future studies.

Reviewer #2 (Remarks to the Author):

Based on additional experiment results and analysis, the reviewer agrees the quality of the whole paper further improved. Many queries from three independent reviewers have been answered in different degree. The authors' great effort is praisable, but there are still some mistakes or unclear points that should be carefully revised before a final acceptance for publication.

We appreciate the reviewers for acknowledging our efforts and providing thoughtful feedback. We have corrected missing parts in the revised manuscript.

1. For glass study, the authors should pay particular attention to thermal analysis like TGA and DSC. However, mistakes still exist for this revised version. 1) In Fig. S1a, the starting point for the TG curves should be 100%. Similar problem was found in Figure S17 for c-Mn-adp.

(Response) We appreciate the reviewer for identifying an oversight on our part. We have adjusted the temperature display range of the graphs, in Supplementary Fig. 1 and 18, ensuring that the starting point of the TGA graphs now begins at 100%.

- Change made:

Supplementary Fig. 1 TGA data of **a**, C-Mg-adp **b**, G-Mg-adp with 10 °C/min ramp rate. T_d was evaluated at 320 °C, which showed abrupt weight loss on TGA.

Supplementary Fig. 18 TGA data of C-M-adp (M = Mn²⁺, Co²⁺, Tb³⁺). T_m and T_d of C-Mn-adp were indicate in the figure.

2) In Fig. S1b, the TG curves for G-Mg-adp is in correct. There is a strange and large weight loss about 20% below 200 oC, but we all know there is no existed solvent for the quenched sample from high-temperature melt.

(Response) We appreciate the reviewer's comments. Previously, the sample was stored in the air, especially during the potentially very humid summer months in our country, for an extended period before the measurement. This extended exposure may have resulted in the surface moisturization of G-Mg-adp. Due to this sample handling, a weight loss occurred at the beginning of the TGA on G-Mg-adp. We acknowledge that, as the reviewer pointed out, these factors could cause confusion for readers. Thus, we carefully conducted a new set of TGA measurements for G-Mg-adp in the as-quenched state.

- Change made: Supplementary Fig. 1b has been replaced with new TGA data using the as-quenched G-Mg-adp sample. Please refer to the modified Supplementary Fig. 1 in the previous response.

3) For the same sample and repeated measurements, why there is such large shift in DSC curves for G-Mg-adp and G-Mn-adp and? See Fig. S30.

(Response) We appreciate the reviewer's comments. In the previous Supplementary Fig. 30, the spacing between each DSC step was increased arbitrarily for improved readability. However, we recognized that presenting information in this manner could be confusing to readers. We have modified the figures using the raw DSC data without any adjustments. The baseline shift toward the exothermic and endothermic directions is attributed to weight loss occurring in the sample and a reduction in the density of the sample, respectively. (*Phys. Rev. Lett.* **112**, 025502 (2014).)

- Change made:

Supplementary Fig. 31 DSC curves of **a**, *G*-Mg-adp and **b**, *G*-Mn-adp from nine consecutive heating and cooling cycles. Only the heating step data is displayed to highlight T_g . Measurements were conducted from 50 °C to slightly below each MOF's T_m , with a heating/cooling rate of 10 °C/min.

4) All the DSC should list the direction of endothermal or exothermal.

(Response) We appreciate the reviewer for pointing out the omission in the heat direction of DSC data. All data were drawn endo-down.

- Change made: The direction of heat flow is now included in Supplementary Fig. 2, 20 and 31.

2. The statement of water stability in Figure S29 is incorrect. A MOF can not be dissolved in solution, and the exact phenomena for C-Mg-adp is decomposition.

(Response) We appreciate and acknowledge the reviewer's comments. To avoid reader confusion, the behavior of C-Mg-adp in water is clearly defined as 'degradation'.

- Change made: Terminology errors on water stability part have been corrected in the main text and supplementary data.

Supplementary Fig. 30 Water stability test of C-Mg-adp and G-Mg-adp. **a**, Photographs for the water treatment processes for C-Mg-adp and G-Mg-adp powders. Notably, G-Mg-adp remains stable even in a large quantity of water, whereas C-Mg-adp undergoes degradation in a small amount of water. **b**, FT-IR data of C-Mg-adp, G-Mg-adp, G-Mg-adp-h, and activated G-Mg-adp-h. **c**, XRPD data of G-Mg-adp, G-Mg-adp-h, and activated G-Mg-adp-h.

“While C-Mg-adp rapidly degrades in a small amount of water, G-Mg-adp almost retains its shape and remains immersed even with an excess quantity of water. As shown in the IR spectra, after soaking in water, the filtered G-Mg-adp (G-Mg-adp-h) retains the coordination bond between Mg²⁺ and carboxylate and reverts back after activation at 80 °C in a vacuum. Throughout the entire

process, *G*-Mg-adp consistently maintains an amorphous states and does not undergo crystallization. The origin of the improved water stability in MOF through vitrification may be revealed through detailed structural analysis of *G*-Mg-adp, *G*-Mg-adp-*h*, and activated *G*-Mg-adp-*h*. However, a complete survey of these topics is beyond the scope of this paper.”

3. Considering the new EXAFS, PDF and water solubility data, it point to a clear possibility that Glassy state has larger Mg cluster compare to crystal state. Could the authors provide ICP and Elemental analysis data to see whether the linker ratio has changed?

(Response) We appreciate the reviewer's comments and suggestions. Because combining the results from two different methods, ICP and EA analysis, can sometimes result in low reliability, we attempted to derive information on Mg and ligand contents from a single measurement, TGA analysis.

Review Only Figure R1. **a**, XPRD data of Mg-adp sample obtained after TGA measurement until 700 °C and a simulated pattern of MgO (COD no. 1011116). **b**, Comparison of weight loss in *C*- and *G*-Mg-adp from the weight at 300 °C with TGA traces. These datasets in this figure are identical to those in Supplementary Fig. 1.

We compared the Mg content between crystal and glass states by assessing the weight of remaining magnesium oxide through TGA. The weight percent of MgO in *C*-Mg-adp and *G*-Mg-adp were assessed based on the weight at 300 °C, a temperature at which the solvents were sufficiently vaporized, serving as the reference point (wt% = 100). As the measurement atmosphere is protected by inert gas, the oxygen species in MgO originates solely from the adipate ligand.

Considering this, the weight loss percentage should be calculated with one oxygen excluded from adipate. The Mg content of samples can be summarized as shown in the table below.

Supplementary Table 5. Metal-ligand ratio of C- and G-Mg-adp calculated from TGA data. The weight of each sample at 300 °C is set as 100 wt% because all guests have been removed. These datasets in the figure below are identical to those in Supplementary Fig. 1.

	MgO wt% ^a	adp - O wt% ^b	Mg mol%/g ^c	adp mol%/g ^d	Mg:adp ratio
G-Mg-adp	24.5 %	75.5 %	0.608 %	0.589 %	1:0.969
C-Mg-adp	23.3 %	76.7 %	0.578 %	0.599 %	1:1.036

^a Remaining weight of the samples heated up to 700 °C

^b Loss in weight of the samples heated up to 700 °C

^c Calculated from MgO wt% using the MW of MgO (40.3 g/mol).

^d Calculated from Adp - O wt% using the MW of adipic acid excluding H₂O (128.1 g/mol).

Given that the formula of C-Mg-adp is Mg₄(adipate)₄(DMA)(H₂O), the results of table are acceptable as the ratio of Mg to adipate is almost 1:1. Overall, the magnesium content of G-Mg-adp is expected to slightly higher than that of C-Mg-adp.

4. Related to previous question, can the glassy MOF be restored to crystalline phase in any method (heating, solvent annealing, etc)? It also shows whether the metal cluster has fundamentally changed.

(Response) We appreciate the reviewer's suggestions. Unfortunately, both Mg-adp and Mn-adp exhibit high glass forming ability (GFA, T_g/T_m) with values of 0.93 and 0.88, respectively. Consequently, they are resistant to crystallization through cooling or heating. This characteristic is also observed in ZIF-62, which, due to its high GFA (0.84), (*Sci. Adv.* **4**, eaao6827 (2018).) remains in a glassy state over the observable time scales. Recrystallization of molten CP/MOFs is primarily

reported in coordination polymers with low GFA, such as Zn-phosphate-azoles and metal-bis(acetamide) networks (see Supplement Fig. 28). Furthermore, *G*-Mg-adp maintains its amorphous nature even when immersed in water or activated after immersion. Therefore, it is difficult to induce recrystallization of *G*-Mg-adp.

5. Though resolution of lab-scale equipment is limited, the raw result of middle and long range correlation should be presented (maybe in SI) rather than deleted. (Response) We appreciate the reviewer's comments. We have included PDF data, which contain middle- and long-range correlations, in the Supplementary Data.

- Change made:

Supplementary Fig. 12 Measured pair distribution function data for *C*-Mg-adp and *G*-Mg-adp with short-, middle- and long-range correlations shown.

6. As the reviewer 1 pointed out, the paper 10.1021/jacs.2c04532, 10.26434/chemrxiv-2023-05kv3, have reported that MOFs composed of ligands with one carboxylate group transform to a glassy state, through a perturbation method. But the authors didn't mention or cite this work. I think a proper way is adding essential discussion and comparison in the introduction part, some as to clearly show the history of carboxylate-based MOF glass and highlight the new advance of presented work.

(Response) We appreciate the reviewer's suggestions for improving the introduction section. We have included the first article mentioned by reviewer as an additional reference (Ref. 24). The second article was recently published in a journal and was cited as References 26 in our previous manuscript. We have also revised the manuscript to more clearly refer to existing research on carboxylate-based MOF glasses instead of the vague word we previously used to represent these studies.

- Change made: However, these studies differ from the present work. The starting materials for glasses in these studies are derived from hydrogen-bonded networks of the metal

complexes or from disorder-induced frameworks created through dehydration. However, these studies differ from the present work in that the starting materials for glasses are ~~hydrogen-bonded networks of the metal complexes or coordination compounds.~~ Certainly, adopting these approaches is a compelling strategy for synthesizing MOF glasses.

Reviewers' Comments:

Reviewer #1:

Remarks to the Author:

The authors have added additional experiments and discussions and have done their best to explain the interesting mechanical properties noted by the reviewers. The new version of the manuscript is more convincing and provides a novel and scientifically high quality argument that is acceptable for acceptance in this journal.

The authors' response to reviewer 1, 1-(3), is interesting in that the relevance of polymer physics to (non-porous) MOFs could be introduced. We urge the authors to cite some relevant papers in this regard and to inspire the wide readership.

Reviewer #2:

Remarks to the Author:

After reading the whole manuscript, this revised version has addressed my previous queries based on supplementary experiments and analysis of all the observed results. The paper quality has also been certainly improved after adopt the kind suggestions from others reviewers. I can now suggest it is suitable for publication.